

# Severe winter haze days in the Beijing-Tianjin-Hebei region from 1985-2017 and the roles of anthropogenic emissions and meteorological parameters

Ruijun Dang[1,2], Hong Liao[3*]

[1]State Key Laboratory of Atmospheric Boundary Layer Physics and Atmospheric Chemistry (LAPC), Institute of Atmospheric Physics, Chinese Academy of Sciences, Beijing, 10029, China
[2]University of Chinese Academy of Sciences, Beijing, 10049, China
[3]Collaborative Innovation Center of Atmospheric Environment and Equipment Technology/Joint International Research Laboratory of Climate and Environment Change, School of Environmental Science and Engineering, Nanjing University of
10 Information Science and Technology, Nanjing, 210044, China

*Correspondence to*: Hong Liao (hongliao@nuist.edu.cn)

**Abstract:** We applied a global 3-D chemical transport model (GEOS-Chem) to examine the variations in the frequency and intensity in severe winter haze days (SWHDs) in BTH from 1985-2017 and quantified the roles of changes in anthropogenic emissions and/or meteorological parameters. Comparisons between the simulated SWHDs and those obtained from the
15 observed $PM_{2.5}$ concentrations and atmospheric visibility showed that the model can capture the spatial and temporal variations of SWHDs in China; the correlation coefficient between the simulated and observed SWHDs is 0.98 at 161 grids in China. From 1985-2017, with changes in both anthropogenic emissions and meteorological parameters, the simulated frequency (total severe haze days in winter) and intensity ($PM_{2.5}$ concentration averaged over severe haze days in winter) of SWHDs in BTH showed increasing trends of 4.5 days decade$^{-1}$ and 13.7 μg m$^{-3}$ decade$^{-1}$, respectively. The simulated
frequency exhibited fluctuations from 1985-2017, with a sudden decrease from 1992-2001 (29 days to 10 days) and a rapid growth from 2003-2012 (16 days to 47 days). The sensitivity simulations indicated that variations in meteorological parameters played a dominant role during 1992-2001, while variations in both emissions and meteorological parameters were important for the simulated frequency trend during 2003-2012 (simulated trends were 27.3 days decade$^{-1}$ and 12.5 days decade$^{-1}$ owing to changes in emissions alone and changes in meteorology alone, respectively). The simulated intensity
showed a steady increase from 1985-2017, which was driven by changes in both emissions and meteorology. The results of this study have important implications for the control of SWHDs in BTH.

## 1 Introduction

Due to rapid industrialization and urbanization, China has suffered from frequent severe haze episodes in recent years (Cai et
al., 2017; Li et al., 2018c; Zhu et al., 2018), especially during winter (Chen and Wang, 2015; Wu et al., 2017; Wang et al.,



2018). The Beijing-Tianjin-Hebei (BTH) region, which is one of the most densely populated economic zones in China, has considerably high level of $PM_{2.5}$ (Jiang, 2015; Chen and Wang, 2015). During severe winter haze days (SWHDs), the detected daily mean $PM_{2.5}$ concentrations reached 300-600 μg m$^{-3}$ in BTH (Wang et al., 2014a; Li et al., 2018b). Exposure to these fine particulate matters detrimentally affects human health, causing problems from respiratory illnesses to

cardiovascular diseases and premature death (Lelieveld et al., 2015; Zhang et al., 2017). High $PM_{2.5}$ levels during SWHDs also severely deteriorate ambient visibility, which endangers ground and air traffic and consequently disrupts economic activities (Wang et al., 2015; Gao et al., 2015a). Therefore, understanding the long-term variations in SWHDs is essential for air quality planning.

Previous studies on severe winter haze were mainly focused on a single episode, which reported the physical and chemical

processes that led to SWHDs. Severe winter haze episodes usually occur under stagnant weather conditions and high anthropogenic emissions (Zhang et al., 2014; Zhang et al., 2015b; Li et al., 2017a). Stagnant weather conditions, characterized by weak surface wind speed, enhanced temperature inversion in the lower troposphere, and depressed planetary boundary layer height (PBLH) restrain the dispersion of pollutants (Liu et al., 2013; Zhao et al., 2013b). High relative humidity (RH) during SWHDs facilitates the rapid growth of secondary aerosols (Sun et al., 2014; Zhang et al.,

2015b). Regional transport from upwind areas ( Zheng et al., 2015b; Sun et al., 2016; Ma et al., 2017) and the feedbacks between aerosols and meteorological variables are also important drivers for the formation of severe winter haze events (Gao et al., 2015b; Zhang et al., 2015a; Tie et al., 2017). The changes in large-scale circulation, such as the anomalous eastward extension of the Siberian high (Jia et al., 2015), weakened East Asian winter monsoon (Li et al., 2016b), and anomalous warm southerlies at 850 hPa (Cai et al., 2017) were also found to be conducive to the formation and maintenance of severe

winter haze in BTH.

Some studies examined the changes in air quality in China during the past decades, but these studies were generally focused on the monthly or seasonal mean haze days. Because of the lack of long-term measurements of $PM_{2.5}$ concentrations in China, observed atmospheric visibility was often used as a proxy for $PM_{2.5}$. Haze days were defined in previous studies as the days with atmospheric visibility of less than 10 km and relative humidity of less than 90 % (Chang et al., 2009; Yang et

al., 2016). Ding and Liu (2014) examined haze days at 553 sites in China and reported that annual haze days averaged over these sites increased from 2-4 days in the 1960s to 11-16 days after 2005. It was also reported that haze days mainly increased in economically developed eastern China and decreased in northeastern and northwestern China. Yang et al. (2016), using visibility data, found that the winter haze days averaged over eastern China (196 sites, 105-122.5° E, 20-45° N) showed a large increasing trend of 2.6 days decade$^{-1}$ during 1980-2012. Historical changes in air quality in China were

also examined through modeling studies. Using the GEOS-Chem model, Jeong and Park (2013) reported a 73 % increase in simulated DJF mean $SO_4^{2-}$-$NO_3^-$-$NH_4^+$ (SNA) concentration between 1985-1989 and 2002-2006 over East Asia (90-145° E, 20-45° N) and largely attributed this increase to changes in anthropogenic emissions. Also by using the GEOS-Chem, Yang et al. (2016) obtained an increasing trend in wintertime $PM_{2.5}$ concentration of 10.5 (±1.5) μg m$^{-3}$ decade$^{-1}$ averaged over




eastern China (105-122.5 °E, 20-45° N) during 1985-2005 and found that the changes in anthropogenic emissions and meteorological parameters over the studied period contributed 83 % and 17 % to the increasing trend in $PM_{2.5}$, respectively.

To date, only a few studies have examined historical changes in SWHDs in China using observed atmospheric visibility. Fu et al. (2013) defined an extremely poor visibility event as a case with atmospheric visibility of less than 5 km for at least 3

5   days and found that the annual frequency of such event averaged over large cites in China (142 cities with populations of more than one million) increased from 1960 to 2009 with a trend of 0.68 events decade$^{-1}$. Using the visibility-converted daily aerosol extinction coefficient (AEC) dataset, Li et al. (2018a) examined the long-term trends in extremely high AEC (the 95$^{th}$ percentile in a year) at 272 sites in China and compared these trends with the trends in median AEC (the 50$^{th}$ percentile in a year). They showed that the extreme trends exceeded the median trends at most sites in the 1980s, the extreme trends were

larger than the median trends in the south (south of 33° N) but smaller in the north (north of 33° N) in the 1990s, and then, most sites exhibited a faster increase in the median trends in the 2000s. These studies, performed through statistical methods based on visibility datasets, could not provide information about the historical changes in aerosol components in SWHDs and associated chemical/physical processes (transport or diffusion, chemical production or loss, wet and dry deposition). Additionally, no quantitative information has ever been given regarding the separate roles of changes in anthropogenic

emissions and meteorological parameters in historical changes of SWHDs in China.

In this study, we applied the GEOS-Chem model to simulate SWHDs over China for 33 winters from 1985 to 2017 and to quantify the roles of changes in anthropogenic emissions and meteorological parameters in the variations of simulated SWHDs over BTH. The observed datasets, GEOS-Chem model and numerical experiments, and the definition of SWHDs are described in Sect. 2. In Sect. 3, simulated $PM_{2.5}$ and SWHDs as well as a model evaluation are presented. In Sect. 4, the

simulated changes in SWHDs over China and the BTH region from 1985-2015 are presented. The key processes that led to SWHDs in BTH are identified by process analysis in Sect. 5. In Sect. 6, the relative roles of changes in anthropogenic emissions and meteorological parameters in SWHDs in BTH are investigated over the studied period.

## 2 Methods

### 2.1 Observed $PM_{2.5}$ concentrations

Hourly surface $PM_{2.5}$ measurements were mainly obtained from the China National Environmental Monitoring Centre (CNEMC) (http://www.cnemc.cn). The monitoring network expanded from 670 sites in 2013 to 1600 sites in 2018, covering approximately 370 cities nationwide. The 670 sites set up in 2013 have obtained continuous measurements since 2013, and 79 sites are located in BTH. Daily mean $PM_{2.5}$ from these 670 sites are used in this study and averaged to the 0.5° latitude x 0.625° longitude MERRA-2 grid (corresponding to 161 model grids) to examine observed SWHDs and for model evaluation.

For the BTH region, hourly $PM_{2.5}$ concentrations that were observed at the U.S. Embassy in Beijing (http://www.stateair.net/web/post/1/1.html) are also used, which are available from 2009 to June 2017.



## 2.2 Observed visibility

Observed atmospheric visibility data were obtained from the National Climatic Data Center (NCDC) Global Summary of Day (GSOD) database (https://www7.ncdc.noaa.gov/CDO/cdoselect.cmd), which has been widely used in previous studies to examine the haze pollution trend in China (Che et al., 2007; Chang et al., 2009; Deng et al., 2012; Yang et al., 2016; Li et al., 2016a). The database includes daily observations from 1985-2018 at 346 sites in China. The Beijing site is located at 39.56°N and 116.17°E.

## 2.3 GEOS-Chem model and numerical experiments

### 2.3.1 Model description

We simulated $PM_{2.5}$ concentrations using the GEOS-Chem model (version 11-01; http://acmg.seas.harvard.edu/geos/) driven by MERRA-2 assimilated meteorological data (Gelaro et al., 2017) from NASA's Global Modeling and Assimilation Office (GMAO). The nested-grid capacity of GEOS-Chem over Asia (11° S-55° N, 60°-150° E) was used with a horizontal resolution of 0.5° latitude by 0.625° longitude and 47 vertical layers up to 0.01 hPa. Boundary conditions for gases and aerosols were updated every 3 h from the coupled global GEOS-Chem simulations performed at a 2°x2.5° horizontal resolution.

The GEOS-Chem model includes fully coupled $O_3$-$NO_x$-hydrocarbon and aerosol chemistry with more than 80 species and 300 reactions (Bey et al., 2001; Park et al., 2004). The $PM_{2.5}$ components simulated in GEOS-Chem include sulfate (Park et al., 2004), nitrate (Pye et al., 2009), ammonium, black carbon and primary organic carbon (Park et al., 2003), mineral dust (Fairlie et al., 2007), and sea salt (Alexander et al. (2005). Aerosol thermodynamic equilibrium is computed by the ISORROPIA II package, which calculates the gas-aerosol partitioning of the sulfate-nitrate-ammonium system (Fountoukis and Nenes, 2007). Heterogeneous reactions of aerosols include irreversible absorption of $NO_3$ and $NO_2$ on wet aerosols (Jacob, 2000), hydrolysis of $N_2O_5$ (Evans and Jacob, 2005), and the uptake of $HO_2$ by aerosols (Thornton et al., 2008). Wet deposition, including below-cloud washout, in-cloud rainout and scavenging in moist convective updrafts follows the scheme of Liu et al. (2001). Dry deposition is computed based on the resistance-in-series model from Wesely (1989) with a number of modifications (Wang et al., 1998). Considering that sea salt is not a dominant aerosol species in China and that the concentrations of mineral dust aerosols are generally low in winter (Duan et al., 2006; Zhao et al., 2013a), we calculated the $PM_{2.5}$ concentration in this study as the sum of the simulated mass of sulfate, nitrate, ammonium, BC, and OC.

### 2.3.2 Emissions

For anthropogenic emissions, global $SO_2$, $NO_x$, $NH_3$, and CO emissions followed the EDGAR v4.2 inventory (EC-JRC/PBL, 2011, http://edgar.jrc.ec.europa.eu/overview.php?v=42). Global BC and OC emissions were taken from the BOND inventory (Bond et al., 2007). Global NMVOC emissions followed the RETRO inventory (ftp://ftp.retro.enes.org/pub/emissions/), where $C_2H_6$ and $C_3H_8$ were replaced with emissions from Xiao et al. (2008). In the East Asian domain, MIX 2010 (Li et al.,



2017b) was the baseline anthropogenic inventory, and annual scaling factors were applied for other years during 1985-2017. Over 1985-2009, scaling factors for all species were derived from the EDGAR v4.3 inventory (http://edgar.jrc.ec.europa.eu/overview.php?v=431), except those for $SO_2$ during 1996-1999 were taken from Lu et al. (2011). For 2011-2017, scaling factors for all species followed the study of Zheng et al. (2018). Global biomass burning emissions were taken from GFED4 for 1997-2016 with a monthly temporal resolution (van der Werf et al., 2017). For the years out of the available range, emissions of the closest available year were used. For example, the biomass burning emissions of 1997 were used for 1985-1996.

Figure 1 shows the variations in wintertime emissions (both anthropogenic and natural emissions) of aerosols and their precursors over eastern China (105-122.5° E, 20-45° N) from 1985-2017. For the $SO_2$, $NO_x$, BC and OC emissions, all are a two-peak type with one peak occurring in the 1990s and another in approximately 2010 (2005 for $SO_2$). The relatively low emissions around 2000 were due to a slowdown in economic growth and a decline in fuel consumption (Streets et al., 2000; Hao et al., 2002; Streets and Aunan, 2005; Ohara et al., 2007). Emissions exhibited drastic increases in the beginning of the 21[st] century and decreases in recent years as a result of the strictly enforced emission reduction measures during the 11[th] and 12[th] Five-Year Plan in China (Zheng et al., 2018). The $NH_3$ emissions continued to increase from 1985-2017 because of the steady growth in agricultural sources. Overall, relative to 1985, $SO_2$, $NO_x$, $NH_3$, BC and OC emissions in 2017 changed by 11 %, 88 %, 81 %, 18 %, and -24 %, respectively. The variations in emissions used in this work are consistent with those from the studies of Jeong and Park (2013) and Yang et al. (2016).

### 2.3.3 Numerical simulations

Daily $PM_{2.5}$ concentrations for winters from 1985-2017 were simulated using the GEOS-Chem model driven by MERRA-2 meteorological fields. The following three simulations were conducted to examine the changes in SWHDs in China from 1985-2017 and to identify the relative roles of changes in anthropogenic emissions and meteorological parameters. The winter (December, January, February, DJF) of a specific year includes December of this year and January and February of the following year.

1. *CTRL*: the control simulation with variations in both meteorological parameters and anthropogenic emissions from 1985-2017.

2. *EMIS*: the simulation with anthropogenic emissions varied from 1985-2017, while the meteorological fields were fixed at the 1985 levels. The aim of this simulation is to quantify only the impacts of changes in anthropogenic emissions on SWHDs during 1985-2017.

3. *MET*: the simulation with meteorological fields varied from 1985-2017, while anthropogenic emissions were fixed at the 2015 levels. This simulation is set to examine only the impacts of changes in meteorological parameters on SWHDs during 1985-2017.



**2.4 Definition of severe winter haze**

To understand the decadal changes in SWHDs, we rely on the GEOS-Chem simulation and use the observed $PM_{2.5}$ concentration and atmospheric visibility for model evaluation. SWHDs must be defined for each of the datasets of observed $PM_{2.5}$, observed atmospheric visibility, and simulated $PM_{2.5}$. For observed $PM_{2.5}$, following the study of Cai et al. (2017), a

severe winter haze day is identified when the daily mean observed $PM_{2.5}$ concentration exceeds the 150 μg m$^{-3}$ threshold because the Chinese government issues a "red alert" when the $PM_{2.5}$ concentration is forecasted to exceed 150 μg m$^{-3}$ for 72 consecutive hours.

With respect to the definition of a severe winter haze day based on observed atmospheric visibility, a severe haze day is defined as a day with visibility of less than a visibility threshold ($V_t$) and relative humidity of less than 90 %. A statistical

approach is used to obtain $V_t$ for each site of interest: 1) select the time period with available observations of both $PM_{2.5}$ and atmospheric visibility; 2) scatterplot the daily atmospheric visibility ($Vis$) versus daily mean $PM_{2.5}$ concentration ($PM_{2.5\_obs}$) for all samples in the time period selected; and 3) perform an exponential fit as $Vis = C_1 + C_2 * \exp(C_3 * PM_{2.5\_obs})$ and obtain the $V_t$ that corresponds to a $PM_{2.5\_obs}$ of 150 μg m$^{-3}$. Notably, atmospheric visibility was observed manually before 2013 and has been observed automatically since 2013. To account for the discrepancies in data before and after 2013, two

sets of $Vt$ are obtained. Figure S1 provides an example of how to compute $V_t$ for Beijing. Observations of $PM_{2.5}$ and atmospheric visibility are available for 2009-2016. Using the statistical approach described above, $V_t$ values are calculated as 6.5 km over the manual period of 2009-2012 and 4.5 km over the automatic period of 2013-2016. Then, the $V_t$ values obtained from 2009-2012 were used to obtain SWHDs for the manually observed period of 1985-2012.

To identify SWHDs based on simulated $PM_{2.5}$, a severe winter haze day is defined as that with a simulated daily mean $PM_{2.5}$

larger than a threshold concentration ($M_t$). Considering the biases in the model results caused by the representation of chemical and physical processes or the horizontal and vertical resolution in the model, $Ct$ is also obtained from a statistical approach for each of the model grids with corresponding $PM_{2.5}$ observations: 1) select the time period with both observed and simulated $PM_{2.5}$ available; 2) for all samples selected in 1), calculate the mean bias (MB) between the daily simulated $PM_{2.5}$ ($PM_{2.5\_mod}$) and daily mean observed $PM_{2.5}$ ($PM_{2.5\_obs}$) as $MB = \sum_{i=1}^{N}(PM_{2.5\_mod} - PM_{2.5\_obs})/N$, where N is total

number of all samples; and 3) set $M_t$ = 150 μg m$^{-3}$ + MB. Using this approach, $M_t$ is 150 μg m$^{-3}$ if the simulated $PM_{2.5}$ has no bias, and $M_t$ is smaller (larger) than 150 μg m$^{-3}$ if the simulated $PM_{2.5}$ concentrations have low (high) biases. Take the grid at Baoding (115.6° E, 39° N), one of the most polluted cities in BTH, as an example. From 2013-2017, when both simulated and observed $PM_{2.5}$ data are available, the simulated wintertime $PM_{2.5}$ from GEOS-Chem has a mean bias of -28.1 μg m$^{-3}$, and therefore, $M_t$ is 121.9 μg m$^{-3}$. As a result, in Baoding, the number of observed SWHDs is 210 days and that of the

simulated SWHDs is 186 days during 2013-2017. Figure S2 displays the calculated $M_t$ values at 161 grids using the above statistical approach. These values were used to identify the SWHDs at each grid from 1985-2017.



## 3 Simulated severe winter haze and model evaluation

### 3.1 Surface-layer PM$_{2.5}$ concentrations in DJF

Figure 2a presents the simulated seasonal (DJF) mean surface-layer PM$_{2.5}$ concentrations averaged from 2013-2017 over China. These years are chosen depending on the available years with measured PM$_{2.5}$ so that the model results can be

evaluated by using observations. Simulated PM$_{2.5}$ concentrations were high over Eastern China, covering important economic zones including BTH, the Sichuan Basin and the Yangtze River Delta (YRD). Simulated highest concentrations occurred in southern BTH, with seasonal mean PM$_{2.5}$ in the range of 135-151 μg m$^{-3}$. For model evaluation, observed PM$_{2.5}$ concentrations from the CNEMC dataset (averaged into 161 model grids as described in Sect. 2.1) are shown in Fig. 2a. The simulated spatial patterns of PM$_{2.5}$ agree well with the measurements, with a high correlation coefficient ($R$) of 0.82 between

the observed and simulated seasonal mean PM$_{2.5}$ values. Figure 2b shows the scatterplot of simulated versus observed seasonal mean PM$_{2.5}$ concentrations at all 161 model grids shown in Fig. 2a. The model overestimates PM$_{2.5}$ concentrations with a normalized mean bias (NMB) of +8.8 % for all grids and an NMB of +4.3 % for BTH.

Figure 3 compares the time series of simulated and observed daily mean surface-layer PM$_{2.5}$ concentrations at the Beijing, Shanghai and Chengdu grids, which represent the three most polluted regions of BTH, YRD and the Sichuan basin,

respectively. For Beijing (116.25° E, 40° N), observations at the U.S. embassy site were used for comparison from 2009-2017. For Shanghai (121.25° E, 31° N) and Chengdu (103.75° E, 30.5° N), observations from the gridded CNEMC dataset were available from 2013-2017. The model generally captures the daily variations (peaks and troughs) in the observed PM$_{2.5}$ concentrations, with R values of 0.61, 0.70 and 0.58 for Beijing, Shanghai and Chengdu, respectively. The model has a low bias in Beijing with an NMB of -9.2 % and is unable to predict the maximum PM$_{2.5}$ concentration in some cases. For

example, observations show that during December 19-26$^{th}$, 2015, there were 7 SWHDs in Beijing, and the highest PM$_{2.5}$ concentration reached 537 μg m$^{-3}$ on December 25. However, the model captured only 3 SWHDs during this period, and the simulated PM$_{2.5}$ concentration was 37 μg m$^{-3}$ on this day. Such under predictions of PM$_{2.5}$ concentrations during severe haze events were also reported by previous studies (Zhang et al., 2015b; Zhang et al., 2014; Wang et al., 2014b) and were attributed to the representation deficiency of the aqueous/heterogeneous processes in the models (Huang, 2014; Zheng et al.,

2015a). For Shanghai and Chengdu, the model has high biases with NMBs of 18.6 % and 28.7 %, respectively.

### 3.2 Frequency and intensity of severe winter haze days

Figure 4a shows the simulated frequencies of SWHDs at 161 model grids averaged from 2013-2017 over China. Consistent with the spatial distribution of seasonal mean PM$_{2.5}$ in Fig. 2a, simulated SWHDs occurred most frequently in BTH. The frequency averaged over BTH was 19.4 days, which is more than twice the mean value of 8.7 days for all 161 grids. The

highest frequencies of 31-39 days were located in southern BTH, indicating that more than one third of the wintertime days experienced severe PM$_{2.5}$ pollution from 2013-2017. For comparison, the observed frequencies of SWHDs, averaged over the same years, are presented in Fig. 4b. The SWHDs were observed most frequently in BTH, and the horizontal





distributions are captured fairly well by the model. The scatterplot (Fig. 4c) of simulated versus observed frequencies of SWHDs for all model grids in Fig. 4a and 4b indicates a high correlation coefficient R of 0.98 and an NMB of -12.3 %. Over BTH, the model underestimated the frequency of SWHDs with an NMB of -8.3 %.

Figure 4d shows the spatial distribution of simulated SWHD intensities. For each grid, the intensity is calculated as the

average of the daily mean $PM_{2.5}$ concentrations over all SWHDs during 2013-2017. The simulated mean $PM_{2.5}$ concentrations of SWHDs were high over the BTH, Shandong and Henan provinces and the Sichuan Basin. The average SWHD $PM_{2.5}$ concentration was 232 μg m$^{-3}$ over BTH, which is much higher than the mean value of 207 μg m$^{-3}$ for all grids. The observed intensities of SWHDs over the same period are also displayed in Fig. 4e. The observed high $PM_{2.5}$ values of SWHDs are centered over BTH, which is generally reflected in the simulation. The model results have low biases in the

Shanxi province and Inner Mongolia and high biases in the Sichuan Basin and Shandong and Henan provinces. Accounting for all model grids in Fig. 4d and 4e, the scatterplot (Fig. 4f) shows that simulated intensities of SWHDs have an NMB of 1.3 %. Over BTH, simulated intensities of SWHDs have a small NMB of 0.3 %.

We also evaluate the model's performance to reproduce the long-term variation in SWHD frequency. To evaluate the historical changes, the SWHDs estimated from the observed atmospheric visibility are used. Beijing is the only grid with

observed $PM_{2.5}$ concentrations available before 2013 for establishing the visibility threshold for identifying visibility-based SWHDs over 1985-2012 (as described in Sect. 2.4). Figure 5 compares the temporal evolution of the SWHD frequency at the Beijing grid obtained from the GEOS-Chem simulation, observed $PM_{2.5}$ (as described in Sect. 3.1), and observed atmospheric visibility (as described in Sect. 2.2). During 2009-2016, when all three datasets are available, frequencies obtained from the simulation and atmospheric visibility capture the interannual variation (peaks and troughs) in observed

frequency well. The model underestimates the frequency with an NMB of -4.9 %, and the visibility result has an NMB of -6.6 %. Over the entire 1985-2017 period, when both observed visibility and simulation are available, the correlation coefficient R is 0.48. The frequency obtained from the atmospheric visibility exhibited a decreasing trend of -4.2 days decade$^{-1}$ during 1985-2002 and an increasing trend of 1.3 days decade$^{-1}$ during 2003-2017; the model partly reproduces these trends with tendencies of -5.2 days decade$^{-1}$ from 1985-2002 and 3.3 days decade$^{-1}$ from 2003-2017. However, a distinct low

bias was found in the simulation before 2003, which may be attributed to the inconsistency problem in the atmospheric visibility dataset obtained from the NCDC. In general, the CTRL simulation can reasonably well capture the spatial distributions and historical changes in SWHD frequencies and intensities in China.

## 4 Simulated decadal changes in severe winter haze days

### 4.1 Decadal changes over China

Figures 6a and 6b show the spatial distributions of simulated SWHD frequencies and intensities averaged from 1985-2017 over China. Simulated SWHDs occurred most frequently in southern BTH, and the intensity showed the highest values in the regions of BTH and Shandong and Henan provinces. For all model grids shown in Fig. 6a, the averaged frequency of



SWHDs was 5.9 days, and the mean intensity was 201 μg m$^{-3}$. Figures 6c and 6d show the simulated temporal changes in SWHD frequency and intensity averaged over all model grids in Fig. 6a from 1985-2017. Both the frequency and intensity of SWHDs exhibited increasing trends over the past three decades. The simulated mean frequency of SWHDs increased from 1.8 days in 1985 to 4.6 days in 2017, with a large growth rate of 2.6 days decade$^{-1}$. The mean SWHD intensity increased from 195 μg m$^{-3}$ in 1985 to 209 μg m$^{-3}$ in 2017, with a linear trend of 6.3 μg m$^{-3}$ decade$^{-1}$.

Over 1985-2017, while the SWHD intensity showed steady growth over the whole period, the changes in SWHD frequency exhibited three distinct stages. The frequency was quite stable, with a value of approximately 3.5 days per winter from 1985-2000, but the value increased rapidly from 2001-2011 and then decreased sharply during 2012-2017; this change can also be seen clearly in Fig. 7, which presents the spatial distributions of linear trends in SWHD frequencies during these three time periods. During 1985-2000, most grids showed no prominent trends, except for some grids in the Henan and Shanxi provinces as well as the Yangtze River Valley, where growth rates were in the range of 0.4-7.5 days decade$^{-1}$. Interestingly, negative trends were found over BTH during this period, with a maximum trend of -6.9 days decade$^{-1}$. From 2001-2011, the SWHD frequencies increased rapidly on a national scale. The largest increasing trends of more than 20 days decade$^{-1}$ were centered on the North China Plain (NCP) and Fen-Wei Plain. Benefiting from the emission reduction actions, the frequencies in most grids began to experience a decline in 2012. Most prominent improvements occurred in the southern BTH and SCB and Shaanxi provinces, with decreasing trends of greater than -45 days decade$^{-1}$.

## 4.2 Decadal changes over BTH

As shown above, BTH merits special attention because the region has high DJF mean PM$_{2.5}$ concentrations, and this region has experienced the most frequently and intensely occurring severe winter haze over the years. To represent the SWHDs over BTH, a regional SWHD is defined here as the day when severe winter haze (defined in Sect. 2.4) occurs simultaneously in more than one third of the grids in a region. Therefore, for the BTH region that include 18 grids, a regional SWHD is identified when 6 grids or more report severe winter haze at a certain time. The intensity of this regional SWHD is calculated as the mean PM$_{2.5}$ concentration of the grids with SWHDs on that day.

Figure 8 shows the temporal evolution of the simulated frequency and intensity, as well as the concentrations of regional SWHD aerosol components in BTH from 1985-2017. Differing from the mean values averaged over 161 grids shown in Fig. 6c, the frequency of regional SWHD in BTH exhibited a much higher level before 2000, indicating serious air pollution by that time. Specifically, the frequency increased from 14 days in 1985 to 29 days in 1992, followed by a sudden decrease to 10 days in 2001, a rapid growth to a peak of 47 days in 2012 and a steep decrease thereafter to 15 days in 2017, which coincides with the temporal variations presented in previous studies for haze days (Chen and Wang, 2015), aerosol extinction coefficient (Li et al., 2016a) and PM$_{2.5}$ concentration (Yang et al., 2018) in the North China Plain (NCP). From 1985-2017, SWHDs in BTH exhibited large increasing trends of 4.5 days decade$^{-1}$ in frequency and 13.7 μg m$^{-3}$ decade$^{-1}$ in intensity. In comparison, the seasonal mean PM$_{2.5}$ concentrations showed a smaller trend of 10.1 μg m$^{-3}$ decade$^{-1}$ (Supplementary Fig. S3), implying a more serious problem with severe winter haze pollution. When considering each



species, nitrate aerosols exhibited the largest increasing trend of 15.7 μg m⁻³ decade⁻¹, which was followed by ammonium (4.5 μg m⁻³ decade⁻¹). Sulfate and BC made little contribution to the increasing trend, while OC showed the opposite trend of -6.8 μg m⁻³ decade⁻¹, which coincided with the emission changes during this period (Fig. 1). Notably, the simulations in this study do not include secondary organic aerosols; therefore, the concentrations of organic aerosols may have been

underestimated. In eastern China, the GEOS-Chem model overestimates nitrate and underestimates sulfate in winter (Wang et al., 2013).

## 5 Key processes for the occurrence of SWHDs over BTH

Concentrations of aerosols are jointly determined by emissions, transport, chemical reactions and deposition. To further isolate and understand the roles of each process in the formation of SWHDs over BTH, a process analysis (PA) was

performed for the CTRL simulation from 1985-2017. PA has been widely used in previous studies to identify the relative importance of atmospheric processes in specific pollution episodes (Goncalves et al., 2009) or annual (Zhang et al., 2009) to decadal simulations (Mu and Liao, 2014; Lou et al., 2015) of air pollutants.

For each aerosol species, six processes were diagnosed at every time step, consisting of primary species emissions, horizontal and vertical transport, chemical reactions, cloud processes, dry deposition, and PBL mixing, which altogether

determined the mass balance of aerosols. Here, the cloud processes include the scavenging of soluble tracers and mixing due to cloud convections. PBL mixing refers to the mass flux brought by turbulence within the planetary boundary layer. PA was carried out for the BTH box (from surface to the 11th vertical model layer that corresponds to approximately 850 hPa) on the basis of daily values. The contribution of each process to the formation of SWHDs was calculated by the following equation:

$$\%PC_i = \frac{PC_{SWH\_i} - PC_{SM\_i}}{\sum_i^n abs(PC_{SWH\_i} - PC_{SM\_i})} * 100 \tag{1}$$

where $n$ is the number of processes ($n$ = 6 as described above), $PC_{SM\_i}$ is the daily mean mass flux of process $i$ from 1985-2017, and $PC_{SWH\_i}$ is the daily mean mass flux of process $i$ averaged over SWHDs during 1985-2017. Note that the sum of $abs(\%PC_i)$ is 100 %.

Figure 9 presents the average daily mass fluxes of PM₂.₅ and its components in the BTH box from each of the six processes over all days during winters from 1985-2017 (stripped bars) and over the SWHDs during this period (solid bars). Averaged

over all days during winters from 1985-2017, the PM₂.₅ in BTH was mainly contributed by chemical reactions (10.1 Gg day⁻¹) and local emissions (5.5 Gg day⁻¹). Net transport had an effect of -12.1 Gg day⁻¹, exporting the particles to the downwind areas as well as to the upper troposphere. The contributions of turbulent diffusion (PBL mixing), dry deposition, and cloud processes were small, with values of -0.8 Gg day⁻¹, -1.8 Gg day⁻¹, and -0.9 Gg day⁻¹, respectively. For all days during winters from 1985-2017, the sum of all processes had a very small value of 0.004 Gg day⁻¹. During SWHDs in the selected time

period, the outflow of particles decreased greatly, from 12.1 Gg day⁻¹ in mean state to 4.8 Gg day⁻¹, which held more particles in BTH. More secondary aerosols were produced through chemical reactions, with an increase from a mean state of



10.0 Gg day$^{-1}$ to 12.0 Gg day$^{-1}$. The suppressed PBL height further restricted the PM$_{2.5}$ diffusion with reduced aerosol outflow from 0.8 Gg day$^{-1}$ to 0.4 Gg day$^{-1}$. Because of the high PM$_{2.5}$ concentrations during the SWHDs, cloud processes and dry deposition removed more aerosols compared to the mean state. Overall, the sum of all processes had a large positive value of 8.1 Gg day$^{-1}$, leading to the accumulation of PM$_{2.5}$ in BTH. The relative contributions of transport, chemistry, cloud processes, dry deposition and PBL mixing to the SWHDs were 65.3 %, 17.6 %, -7.5 %, -6.4 % and 3.2 % (Table 1), respectively, indicating that transport was the most important factor that facilitated the occurrence of SWHDs in BTH. Notably, the relative contribution of emissions is zero because the emissions in the model were monthly values, which were the same for either polluted or clean days. During the SWHDs, the relative contribution of transport is the largest for all aerosol species except sulfate (Table 1). For sulfate aerosols, chemical reactions had a large relative contribution of 49.3 %, which is comparable to the transport contribution (39.5 %). This finding highlighted the importance of local chemical production of sulfate during SWHDs. With regard to the carbonaceous aerosols, transport was the dominant process with a relative contribution of 83.2 % since carbonaceous aerosols are assumed to be chemically inert in the model. For nitrate and ammonium, transport had relative contributions of 62.4 % and 61.3 %, respectively.

To further explore the wind component (zonal, meridional or vertical) that contributed most to the transport process, mass fluxes brought by winds in three directions (east-west, north-south, up-down) were also determined for the BTH box from 1985-2017. Then, relative contributions were calculated following Eq. (1) except that *n* is the number for the three directions. The north-south (NS) wind was calculated to have a relative contribution of 93 % and hence a dominant role (pie chart in Fig. 9). This result agrees with previous studies (Li et al., 2016b; Cai et al., 2017), which found that anomalous southerlies in the lower troposphere favor the formation and accumulation of PM$_{2.5}$ during severe haze events.

## 6 Roles of anthropogenic emissions and meteorological parameters

Figure 10a shows the time series of regional SWHDs frequencies in BTH from 1985-2017 based on three simulations (CTRL, EMIS, and MET). In the CTRL simulation, with changes in both emissions and meteorology, the SWHD frequency in BTH increased from 1985-2017, with a linear trend of 4.5 days decade$^{-1}$ (as described in Sect. 4.2). During 1985-2017, the simulated SWHD frequency exhibited an increasing trend of 5.9 days decade$^{-1}$ in EMIS with changes in emissions alone but no significant trend was seen in the MET simulation with variations in meteorological parameters alone. Considering that the frequency trend is dependent on the selected simulation years, we show in Fig. 10b-10d a more comprehensive analysis of frequency trends over different selected periods, with a minimum duration of 10 years. Here, the *x-axis* indicates the start year, and the *y-axis* indicates the number of years since the start year during which period the trend is calculated. As shown by the trends in the 10 years since the start year, the CTRL frequency experienced a moderate increase in the late 1980s, a significant decrease beginning in 1991, and a large increase beginning in 1997. Similar patterns are also found in the EMIS and MET simulations, although different magnitudes are simulated. During 1992-2001 when the decreasing trend (-20.1 days decade$^{-1}$) was the largest in the CTRL simulation, the trends in EMIS and MET were -3.3 days decade$^{-1}$ and -20.1 days



decade$^{-1}$, respectively, indicating that the variations in meteorological parameters dominated the decreasing trend over this period. The trend of -3.3 days decade$^{-1}$ from 1992-2001 in EMIS was consistent with the emission reductions over this period (Fig. 1). During 2003-2012 when the increasing frequency trend (23.6 days decade$^{-1}$) was the largest in the CTRL simulation, the EMIS and MET trends were 27.3 days decade$^{-1}$ and 12.5 days decade$^{-1,}$ respectively, reflecting that both the

increases in emissions and changes in meteorological parameters contributed to the increasing trend in CTRL, although the role of meteorology was smaller compared to that of emissions. Notably, the decrease in frequency over the 1990s in MET was mainly caused by the interdecadal variations in atmospheric circulations (Chen and Wang, 2015). The researchers found that compared to the meteorological conditions from 1993-2001, those from 1984-1992 and 2002-2010 were more favorable for the occurrence of severe winter haze events over North China because of the weaker wind speeds and increased moisture

in the lower troposphere, weaker East Asian trough in the midtroposphere, and greater northward shift of the East Asian jet stream in the upper troposphere.

Figure 11a shows the time series of the regional SWHD intensities in BTH from 1985-2017 based on the CTRL, EMIS, and MET simulations. From 1985-2015, the simulated SWHD intensity exhibited increasing trends of 13.7 μg m$^{-3}$ decade$^{-1}$, 6.2 μg m$^{-3}$ decade$^{-1}$, and 8.0 μg m$^{-3}$ decade$^{-1}$ due to changes in both emissions and meteorology, changes in emissions alone, and

changes in meteorological parameters alone. Therefore, both anthropogenic emissions and meteorological parameters were major factors that led to high PM$_{2.5}$ concentrations in the SWHDs in BTH. The intensity in CTRL exhibited an overall increasing trend except for a small decrease from 1992-2001 (Fig. 11b), due to the reasons mentioned above.

Additionally, we examine the interannual variations in SWHDs in BTH. The 9-point weighted running mean is employed for the frequencies in Fig. 10a and intensities in Fig. 11a, respectively, to remove the fluctuations in periods of more than 9

20  years while reserving the interannual anomalies. The weighting coefficients are based on the Lanczos filter (Duchon, 1979). Figure 12 shows the interannual variations in frequencies and intensities from the three simulations (CTRL, EMIS, and MET) from 1989-2013. The interannual frequency variation in MET generally resembles that from the CTRL simulation, with a high correlation coefficient R of 0.80 between the simulations, while the R between the interannual variations of CTRL and EMIS is only 0.05 (Fig. 12a). Similarly, for intensity, the interannual variation from the MET exhibited the same peaks and

troughs as that in the CTRL simulation, with a high R of 0.90 between the simulations (Fig. 12b). Therefore, the interannual SWHD variations in BTH were mainly driven by variations in meteorological parameters. To quantify the interannual variation caused by the meteorological variations, the relative interannual change (*%IC*) was calculated by the following equation:

$$\%IC_i = \frac{abs(MET\_IA_i - MET\_IA_{i-1})}{CTL\_F_{i-1}} * 100 \qquad (2)$$

where *MET_IA$_i$* and *MET_IA$_{i-1}$* are the interannual anomalies in the MET simulation in years *i* and *i-1*, and *CTL_F$_{i-1}$* is the original frequency or intensity in the CTRL simulation in year *i-1*. The calculated mean *%IC* was 26 % for frequency from 1989-2013, and 5.3 % for intensity over the same period. Here, the *%IC* indicates that on average, meteorological variations alone can lead to a change of ±26 % in frequency and ±5.3 % in PM$_{2.5}$ concentrations for the SWHDs in BTH interannually.



This highlights the importance of variations in meteorological parameters for policy makers when actions are taken to improve air quality in the short term.

**7 Conclusions**

We investigated the changes in SWHDs (days with observed $PM_{2.5}$ concentrations larger than 150 µg m$^{-3}$) from 1985-2017

and quantified the respective roles of changes in anthropogenic emissions and meteorological parameters in the simulated SWHD changes using the nested-grid version of the GEOS-Chem model.

The simulated SWHDs were evaluated using the observed $PM_{2.5}$ concentrations and atmospheric visibility. The GEOS-Chem model captured the daily variations in $PM_{2.5}$ concentrations fairly well. The correlation coefficient (R) and normalized mean bias (NMB) between the simulated and observed wintertime daily $PM_{2.5}$ concentrations were 0.61 and -9.2 % in Beijing

from 2009-2016, 0.70 and 18.6 % in Shanghai from 2013-2017, and 0.58 and 28.7 % in Chengdu from 2013-2017. The model also captured the SWHD frequency (total severe haze days during winter) and intensity ($PM_{2.5}$ concentration averaged over severe haze days during winter); compared to the observed SWHD frequency and intensity at 161 grids in China from 2013-2017, the simulated frequency had an R of 0.98 and NMB of -12.3 %, and the simulated intensity had an NMB of 1.3 %.

The Beijing-Tianjin-Hebei region (BTH) had the highest SWHD frequency and intensity based on the observed $PM_{2.5}$ concentrations from 2013-2017, with a frequency of 21.2 days yr$^{-1}$ and a mean intensity of 231.6 µg m$^{-3}$. Historically, from 1985-2017, with changes in emissions and meteorology, the simulated SWHD frequency in BTH increased with a linear trend of 4.5 days decade$^{-1}$, and the simulated intensity increased by 13.7 µg m$^{-3}$ decade$^{-1}$, which was larger than the increase in the seasonal mean $PM_{2.5}$ concentration (10.1 µg m$^{-3}$ decade$^{-1}$). Further process analysis was carried out to identify the key

processes for SWHD occurrences in BTH. The comparisons of the six processes (emission, transport, chemistry, cloud processes, dry deposition, and PBL mixing) averaged over the SWHDs with those averaged over all days during winters from 1985-2017 indicate that transport (outflow of particles) had the largest change from -12.1 Gg day$^{-1}$ in the mean state to -4.8 Gg day$^{-1}$ in SWHDs, which had the largest contribution to SWHD formation. Transport, chemistry, cloud processes, dry deposition, and PBL mixing had relative contributions of 65.3 %, 17.6 %, -7.5 %, -6.4 % and 3.2 %, respectively, to SWHD

formation.

From 1985-2017, the simulated frequency of SWHDs in BTH experienced several stages, including an increase from 14 days in 1985 to 29 days in 1992, a sudden drop to 10 days in 2001, a rapid growth to the peak of 47 days in 2012, and a steep decrease thereafter to 15 days in 2017. Sensitivity studies showed that the decrease in frequency from 1992-2001 was mainly caused by changes in meteorological parameters. From 2003-2012, when the SWHD frequency increased sharply, the

simulated frequency trends were 23.6, 27.3, and 12.5 days decade$^{-1}$ owing to the changes in both emissions and meteorology, emissions alone, and meteorology alone, respectively, highlighting the contributions of both emissions and meteorological parameters. Interestingly, the simulated SWHD intensity in BTH increased steadily from 1985-2017; variations in emissions





alone and meteorology alone both enhanced the intensity growth with trends of 6.2 μg m$^{-3}$ decade$^{-1}$ and 8.0 μg m$^{-3}$ decade$^{-1}$, respectively. These results suggest that meteorological parameters must be considered for the control of SWHDs.

Note that a number of factors contribute to the uncertainties in our results. We obtained long-term SWHDs by using the observed atmospheric visibility, which were influenced by the changes in the way of observation or relocation of observation site, especially before 2013 when observations were carried out manually. The uncertainties with emissions inventories of aerosols and aerosol precursors might lead to uncertainties in our simulated decadal changes in aerosol concentrations. Furthermore, we only accounted for anthropogenic aerosols in our simulations; natural aerosols, such as secondary organic aerosol, mineral dust, and biomass burning aerosols, contributed to the observed changes in SWHDs and should be considered for in model simulation. These issues need to be examined in future studies of severe haze trend over eastern China.

**Data availability**

The GEOS-Chem model is an open-access model managed by the Atmospheric Chemistry Modeling group at Harvard University with support from institutes in North America, Europe, and Asia. The source codes can be downloaded from http://acmg.seas.harvard.edu/geos/. The observed daily PM$_{2.5}$ concentrations are derived from the China National Environmental Monitoring Centre (CNEMC; http://www.cnemc.cn) for 670 sites from 2013-2018 and from the U.S. Embassy (http://www.stateair.net/web/post/1/1.html) for the Beijing site from 2009 to June 2017. Observed atmospheric visibility at the Beijing site from 1985-2018 was obtained from the National Climatic Data Center (NCDC) Global Summary of Day (GSOD) database, which is available at https://www7.ncdc.noaa.gov/CDO/cdoselect.cmd.

**Author contributions**

HL and RD conceived of the study and designed the experiments. RD carried out the simulations and performed the analysis. RD and HL prepared the manuscript with contributions from all coauthors.

**Competing interests**

The authors declare that they have no conflicts of interest.

**Acknowledgements**

This work was supported by the National Natural Science Foundation of China under grants 91544219, 91744311, and 41475137. We acknowledge the CNEMC, U.S. Embassy, NCDC, EDGAR and MEP teams for making their data publicly available. We acknowledge the efforts of GEOS-Chem working groups for developing and managing the model.




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



**Tables**

**Table 1.** Relative contributions of each of the five processes (transport, chemistry, cloud, dry, and PBL mixing) to SWHD formation in BTH (unit: %). The process analyses are carried out for both $PM_{2.5}$ and its components, following Eq. (1) described in Sect. 5.

|  | Transport | Chemistry | Cloud | Dry | PBL mixing |
|---|---|---|---|---|---|
| $PM_{2.5}$ | 65.3 | 17.6 | -7.5 | -6.4 | 3.2 |
| Sulfate | 38.5 | 49.3 | -6.8 | -4.0 | 1.4 |
| Nitrate | 62.4 | 18.8 | -9.0 | -7.1 | 2.7 |
| Ammonium | 61.3 | 21.8 | -7.4 | -6.8 | 2.7 |
| Carbon | 83.2 | 0 | -5.8 | -6.2 | 4.8 |



**Figures**

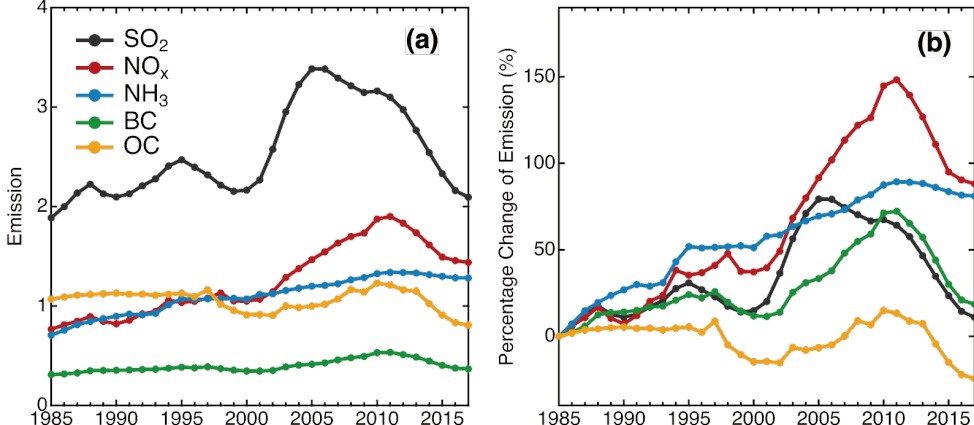

**Figure 1.** (a) Variations in total emissions (anthropogenic plus natural emissions) of sulfur dioxide (SO$_2$, Tg S DJF$^{-1}$), nitrogen oxide (NO$_x$, Tg N DJF$^{-1}$), ammonia (NH$_3$, Tg N DJF$^{-1}$), black carbon (BC, Tg C DJF$^{-1}$) and organic carbon (OC, Tg C DJF$^{-1}$) over eastern China (105-123° E, 20-45° N) during December-January-February (DJF) from 1985-2017 and (b) the percentage changes relative to 1985 values.





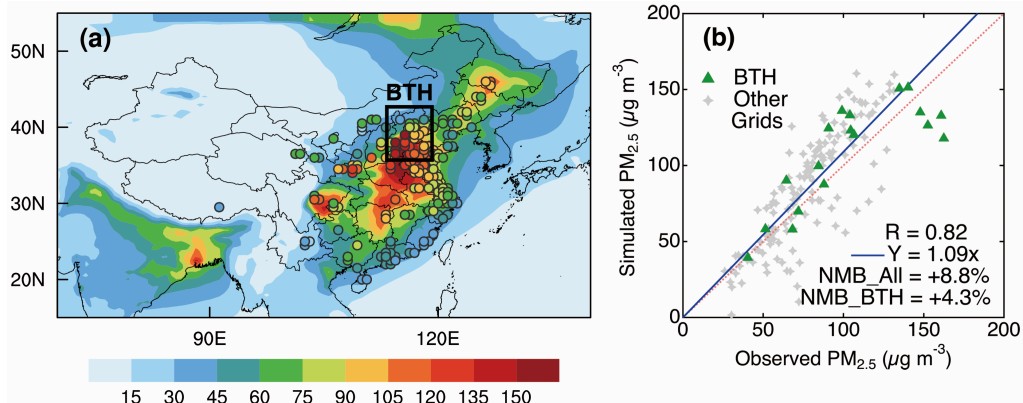

**Figure 2.** (a) Spatial distributions of simulated (CTRL, shades) and observed (CNEMC, dots) seasonal (DJF) mean surface-layer $PM_{2.5}$ concentrations ($\mu g\ m^{-3}$) averaged over 5 winters (2013-2017). (b) Scatterplot of simulated versus observed DJF mean $PM_{2.5}$ concentration ($\mu g\ m^{-3}$) averaged from 2013-2017 for 161 grids in (a), where the green grids are 18 grids located in the BTH (Beijing-Tianjin-Hebei) region. Also shown in (b) are the y=x line (dashed), linear fit (solid line and equation) and values of r and NMB. Here, r is the correlation coefficient between simulated and observed $PM_{2.5}$ concentrations. NMB (Normalized mean bias) = $(\sum_{i=1}^{N}(Mi - Oi) / \sum_{i=1}^{N}(Oi)) \times 100\ \%$, where $Oi$ and $Mi$ are the observed and simulated $PM_{2.5}$ concentrations, respectively. $i$ refers to the $i$th site, and $N$ is the total number of sites.





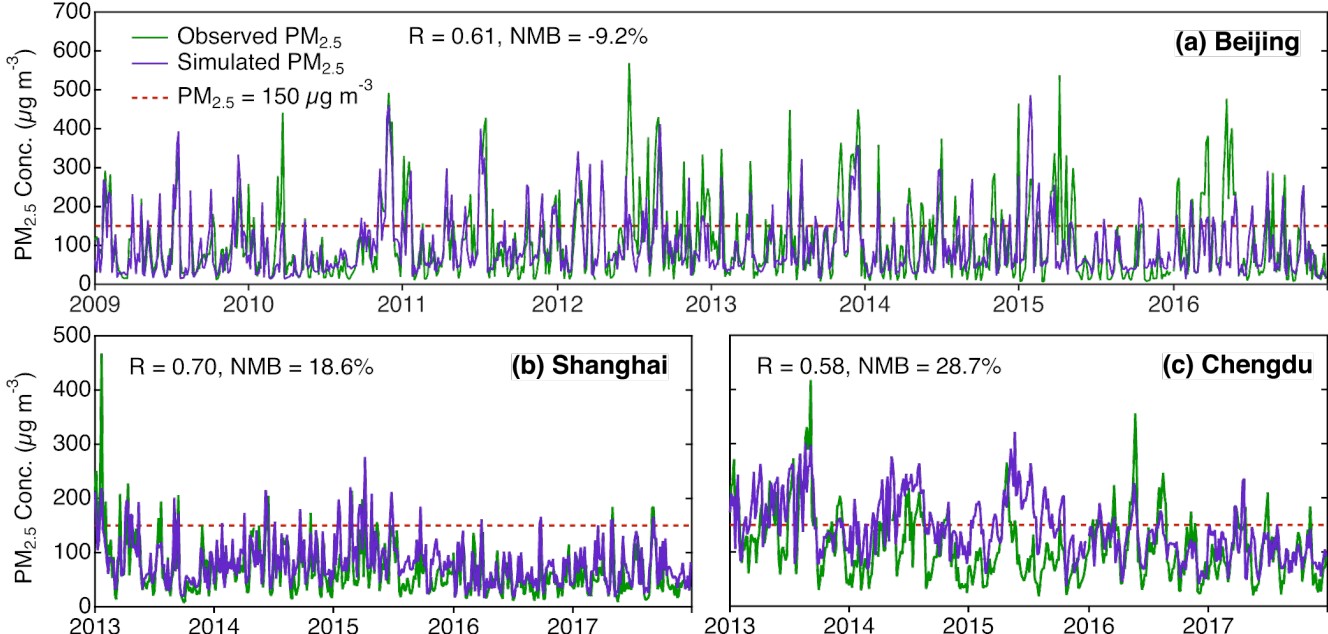

**Figure 3.** Simulated (purple solid line, from CTRL simulation) and observed (green solid line, from the U.S. Embassy and CNEMC) daily mean surface-layer $PM_{2.5}$ concentrations ($\mu g\ m^{-3}$) for grids of (a) Beijing, (b) Shanghai and (c) Chengdu over the period when observations are available. Also shown are the threshold of $PM_{2.5} = 150\ \mu g\ m^{-3}$ (red dashed line), the correlation coefficients (R) and NMB values between observed and simulated daily mean $PM_{2.5}$ concentrations.




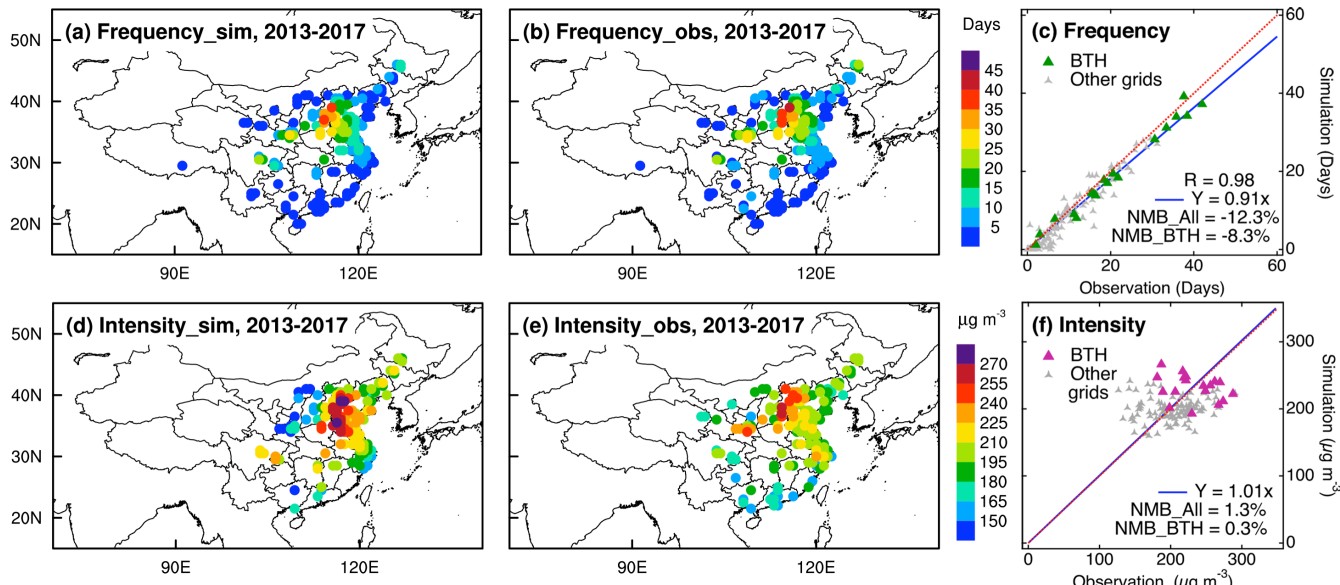

**Figure 4.** Spatial distributions of (a) simulated and (b) observed frequencies (days) of SWHDs averaged from 2013-2017 at 161 model grids in China and (c) scatterplot of simulated versus observed results in (a) and (b); (d-f) are the same as (a-c) except that the values are SWHD intensities (μg m⁻³). Also shown in the scatterplots are y=x lines (dashed), linear fits (solid line and equation), correlation coefficients (R) and NMB values.





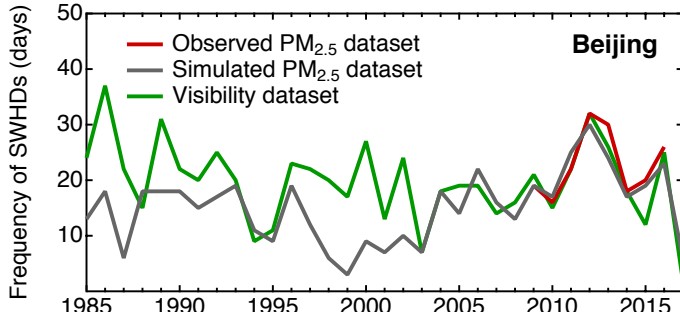

**Figure 5.** Temporal evolutions of SWHD frequency (days per DJF) at the Beijing grid, which were derived from observed $PM_{2.5}$ (red, 2009-2016), simulated $PM_{2.5}$ (gray, 1985-2017) and observed atmospheric visibility (green, 1985-2017).



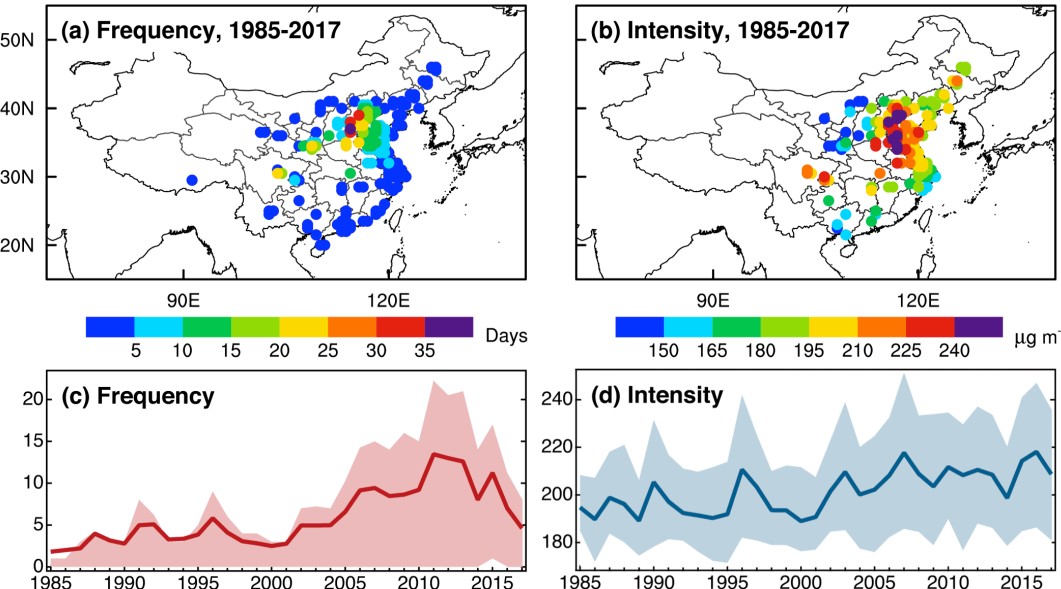

**Figure 6.** Spatial distributions of SWHD (a) frequencies (days) and (b) intensities (µg m⁻³) at 161 model grids averaged from 1985-2017. Temporal evolutions of SWHD (c) frequency (days) and (d) intensity (µg m⁻³) in China from 1985-2017. In (c) and (d), solid lines are the averages of all samples (161 model grids), and the shades indicate the range between the first and third quartiles of the samples.





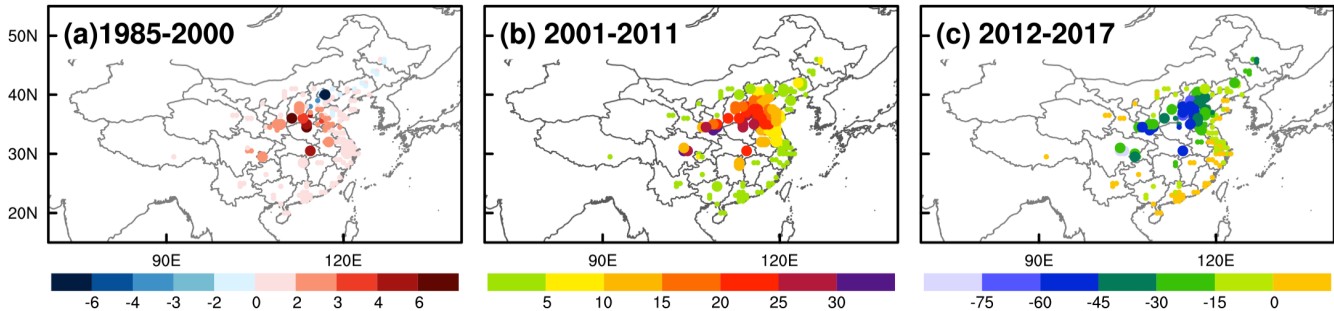

**Figure 7.** Linear trends of simulated frequencies (days decade$^{-1}$) of SWHDs at 161 model grids over the three periods of (a) 1985-2000, (b) 2001-2011 and (c) 2012-2017. Large dots indicate statistical significance above the 95 % confidence level, while smaller ones are not statistically significant.





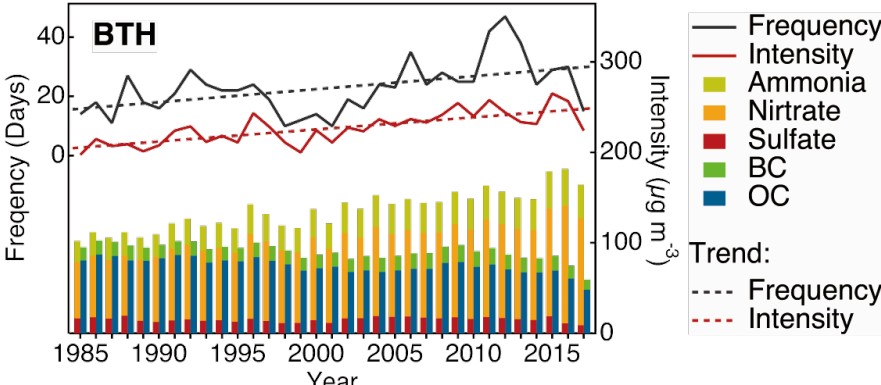

**Figure 8.** Simulated temporal variations in frequency (days, black line), intensity (μg m⁻³, red line) and concentrations of PM₂.₅ components (μg m⁻³, bars): ammonia (blue), nitrate (yellow), sulfate (red), black carbon (purple) and organic aerosols (green) of regional SWHDs in BTH from 1985-2017. Also shown are linear trends (dashed lines) for frequency and intensity, which are statistically significant above the 95 % confidence level.



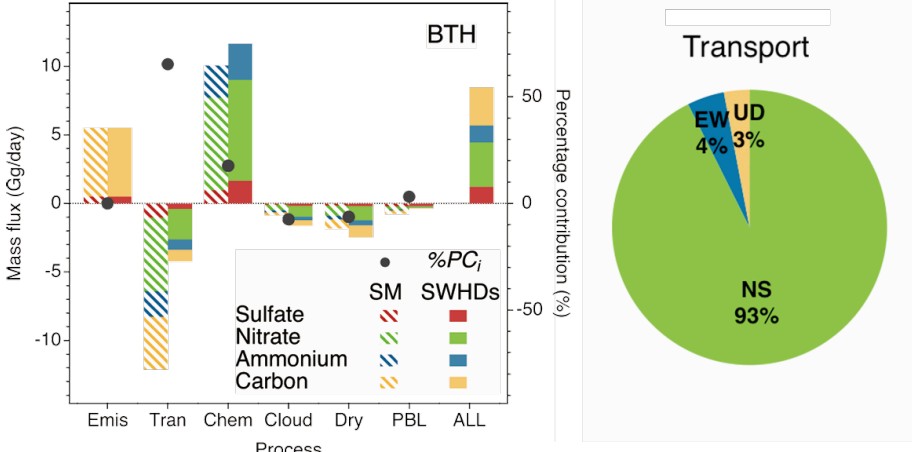

**Figure 9.** Daily mean mass fluxes of aerosols by physical/chemical processes (bottom axis: emission, transport, chemistry, cloud processes, dry deposition, diffusion and the sums) in the BTH box. Two kinds of averages are given here: averaged over all days during winters from 1985-2017 ($PC_{SM}$, left stripped bars) and averaged over all SWHDs during this period ($PC_{SWHD}$, right solid bars). Black dots give the relative contributions (*%PC*, right axis) of each of the six processes $i$ (unit: %) for the SWHD formation when compared with mean values. The *%PC$_i$* is calculated following Eq. (1). The pie chart presents the relative contributions made by 3 winds components: north-south (NS), east-west (EW) and up-down (UD) for the transport difference ($PC_{\_SWHD\_tran} - PC_{\_SM\_tran}$). Note that the carbon here refers to the carbonaceous aerosols, which is the sum of black carbon and organic aerosols.



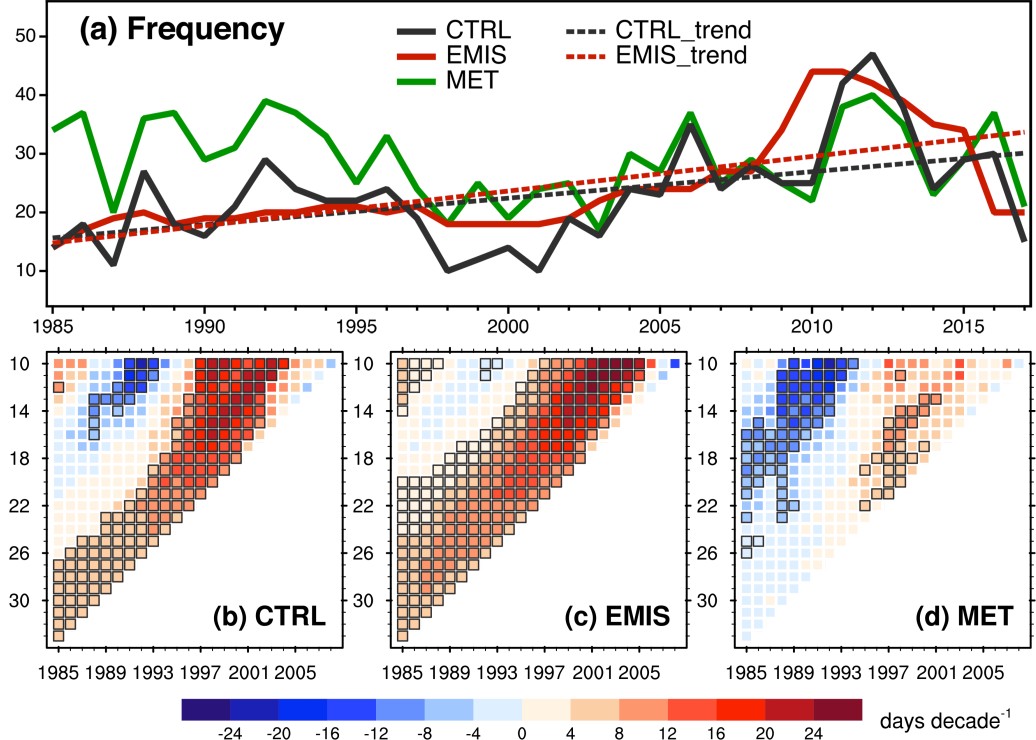

**Figure 10.** (a) Time series of frequencies (days) of regional SWHD in BTH from three simulations (CTRL, EMIS, MET) for 1985-2017. (b-e) Time series of linear trends calculated over different periods for simulated frequencies of the (b) CTRL, (c) EMIS, and (d) MET simulations. The *x-axis* indicates the start year, and the *y*-axis indicates the number of years since the start year during which period the trend is calculated. The filled color in each square shows the calculated trend value, and those values marked with black borders are significant above the 95 % confidence level.



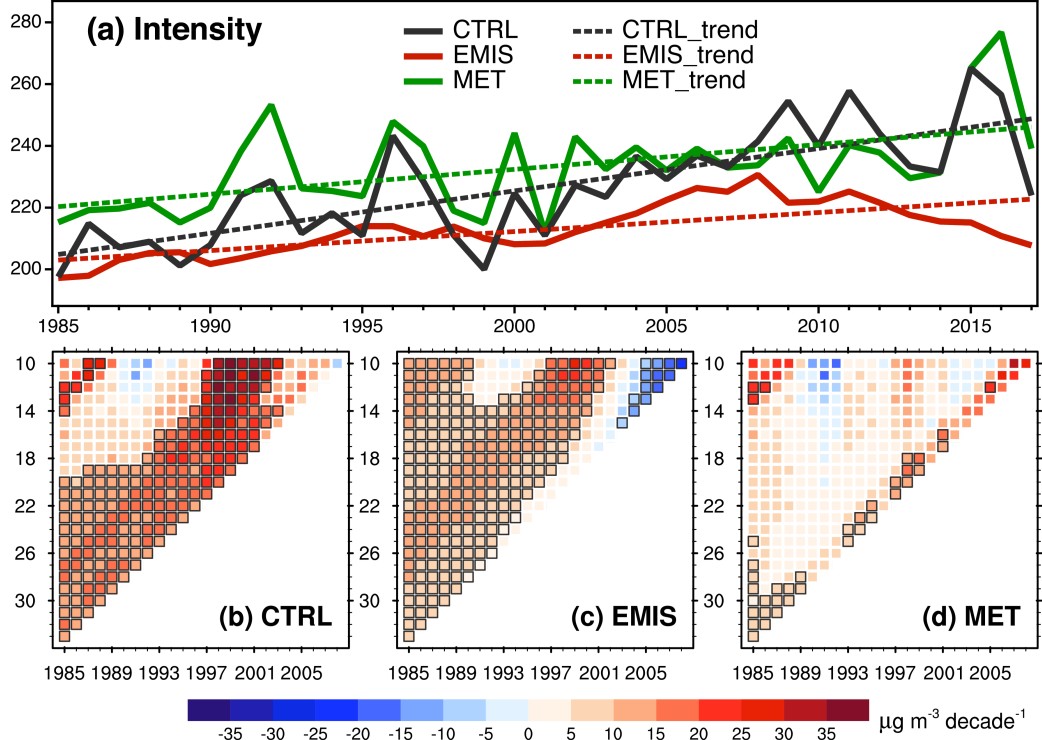

**Figure 11.** (a) Time series of intensities (µg m⁻³) of regional SWHD in BTH from the three simulations (CTRL, EMIS, MET) for 1985-2017. (b-e) Time series of linear trends calculated over different periods for simulated intensities of the (b) CTRL, (c) EMIS, and (d) MET simulations. The *x-axis* indicates the start year, and the *y*-axis indicates the number of years since the start year during which period the trend is calculated. The filled color in each square shows the calculated trend value, and those marked with black borders are significant above the 95 % confidence level.





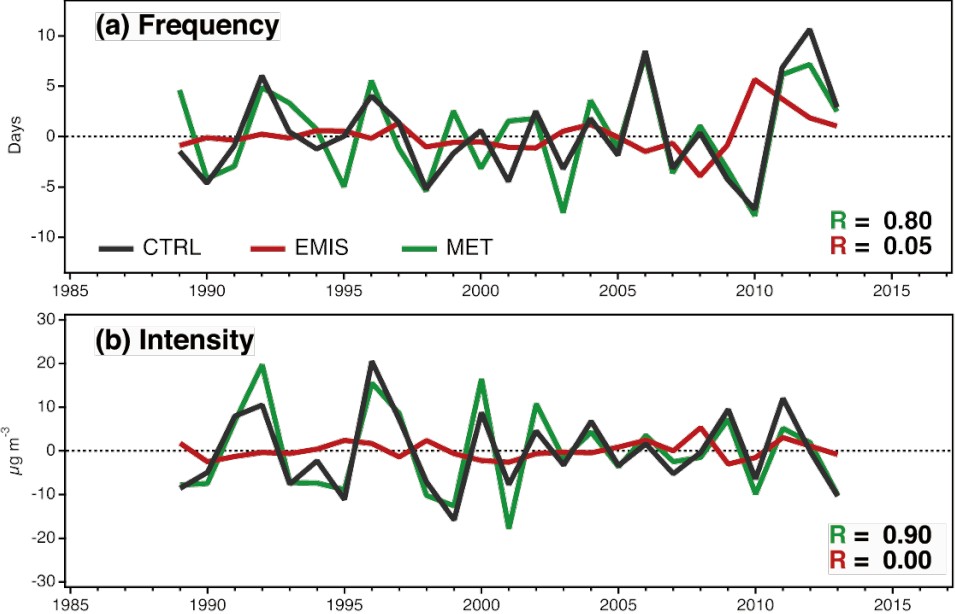

**Figure 12.** Interannual variations in SWHD (a) frequencies (days) and (b) intensities ($\mu$g m$^{-3}$) in BTH through a nine-year weighted moving average method. The 9-point high-pass Lanczos filter was used as the weighting coefficients, which effectively removes fluctuations with periods of more than 9 years and reserves the interannual anomalies. Also shown are the correlation coefficients that were calculated between the interannual signals of the CTRL and MET simulations (green R) and between the results of the CTRL and EMIS simulations (red R).