# Peer review of "Severe winter haze days in the Beijing-Tianjin-Hebei region from 1985-2017 and the roles of anthropogenic emissions and meteorological parameters"

_Atmospheric Chemistry and Physics, 2019_

## Referee Comment (RC1) · Anonymous Referee #1 · 17 May 2019

This paper investigates the roles of meteorology and emission on the long term changes of winter haze in the BTH region. To my knowledge, this is the first attempt to quantify these two effects for historical periods. The results shown here thus have important implications for the understanding of haze formation and air quality control in north China. The paper is also well organized and easy to follow. I only have a few minor comments as listed below:

1. It is better to include some more quantitative results in the abstract, such as those in Table 1 and Figure 12. 2. Section 2.2: Historical visibility data usually have higher

uncertainty and noise level, I wonder if any quality control is enforced? 3. Section 2.3.1: The authors perform nested simulation with high resolution. However, the input meteorology data is still of low resolution. I wonder if this will affect the simulation results? Or is nested simulation really necessary? 4. Section 4.2: I suggest adding some discussion about the uncertainty of the trend of each species, as this can be significantly affected by uncertainties in emission (especially historical emission data) and chemistry processes in the model. 5. Section 5: When using equation (1) to make the partition, the contribution of each factor is assumed to be linear. This may not always be true. For example, both transport and PBL mixing can be affected by horizontal wind speed. So maybe a note is needed here when interpreting the results? 6. Figure 9 is very interesting and important. I wonder if the results are similar for different periods with different haze day trends?

---

## Referee Comment (RC2) · Anonymous Referee #2 · 28 May 2019

Review of "Severe winter haze days in the Beijing-Tianjin-Hebei region from 1985–2017 and the roles of anthropogenic emissions and meteorological parameters" by Dang and Liao

This paper addresses the interannual variation of the severe winter haze days (SWHDs) in the Beijing-Tianjin-Hebei (BTH) region from 1985–2017 and the impacts of the anthropogenic emissions and meteorology on the variation.

This study is of scientific importance and the research is well conducted. The paper is written with clear structure, good illustrations, and convincing discussions. The paper provides an enhanced understanding on this topic. In the meantime, the paper is subject to some issues described in the following. The authors are encouraged to consider these questions in their revision.

The following are some questions for the authors to consider when revising their paper.

Major issues:

1. It appears that GEOS-Chem cannot capture the interannual variation of SWHDs intensity well (see Figure 4f). In Figure 4f, $R$ is missing. Is R statistically significant? How does this uncertainty affect the results in Figures 6, 11 and 12 and the associated discussion in the text?

2. In the simulation experiments, biomass burning emissions are interannually variable from 1997-2016. The interannual variation of biomass burning is large globally and regionally, which may not be ignored. Therefore, the EMIS simulation may be driven by the interannual variations in both anthropogenic and biomass burning emissions, while the MET simulation is influenced by both biomass burning emissions and meteorology. Please explain.

3. In the abstract, the authors stated: "the correlation coefficient between the simulated and observed SWHDs is 0.98 at 161 grids in China". This claim is based on Figure 4 that compares simulated and observed SWHD in terms of frequency and intensity. In section 2.4, the authors defined SWHDs for the observations and simulations. The simulated SWHD is adjusted according to the simulation biases. It is not clear if the simulated results presented in Figure 4 are the original simulations or adjusted values. If they are adjusted values, it should be stated there. Also, the claim in the abstract should be revised accordingly.

4. In generally, a moving average tends to smooth year-to-year variation. But in Figure 12, the author stated that using a "9-year weighted moving average" can reserve interannual variations in the SWDs frequency and intensity. I wonder how this works. Why 9 years? Which climatic influence did the authors try to remove? Why to remove the fluctuations of more than 9 years?

Minor issues:

5. In the title, the term of "meteorological parameters" is not suitable. It is better to use term of "meteorology", "meteorological factors", or "meteorological variables".
6. Figure 8, it is hard to identify a tick for a year in the x-axis.
7. P. 7, Line 32, replace "horizontal" with "spatial".
8. In Author contributions, delete "from all coauthors".

---

## Author Comment (AC1) · 22 Jul 2019

**Response to Comments of Reviewer #1**

**Manuscript number:** acp-2019-306

**Authors:** Ruijun Dang and Hong Liao

**Title:** Severe winter haze days in the Beijing-Tianjin-Hebei region from 1985-2017 and the roles of anthropogenic emissions and meteorology

**General comments:**

*This paper investigates the roles of meteorology and emission on the long-term changes of winter haze in the BTH region. To my knowledge, this is the first attempt to quantify these two effects for historical periods. The results shown here thus have important implications for the understanding of haze formation and air quality control in north China. The paper is also well organized and easy to follow. I only have a few minor comments as listed below.*

**Response:**

Thanks to the reviewer for the helpful comments and suggestions. We have revised the manuscript carefully and the point-to-point responses are listed below.

**Specific Comments:**

*1. It is better to include some more quantitative results in the abstract, such as those in Table 1 and Figure 12.*

**Response:**

We have added the following quantitative descriptions in the abstract to summarize the major results from Table 1 and Figure 12:

"Process analysis on all SWHDs during 1985-2017 indicated that transport was the most important process for the formation of SWHDs in BTH with a relative contribution of 65.3 %, followed by chemistry (17.6%), cloud processes (-7.5%), dry deposition (-6.4%) and PBL mixing (3.2%). Further examination showed that SWHDs exhibited large interannual variations in frequency and intensity, which were mainly driven by changes in meteorology."

*2. Historical visibility data usually have higher uncertainty and noise level, I wonder if any quality control is enforced?*

**Response:**

Yes, quality control was enforced in this work. First, the downloaded daily visibility

data from National Climatic Data Center (NCDC) have already undergone extensive quality control (see ftp://ftp.ncdc.noaa.gov/pub/data/gsod/readme.text and ftp://ftp.ncdc.noaa.gov/pub/data/noaa/ish-qc.pdf for details), which to a large extent have eliminated many random errors in the original data.

Second, to ensure the consistency and continuity of the downloaded data, further quality control was carried out to remove unreliable outliers/stations following the methods in previous studies (Li et al., 2016; He et al., 2017). For each station:

1) If the standardized daily visibility ($V_i$) exceeds ± 4, the daily data is then marked as outlier and removed from the original time series.

2) A 3-year sliding window was applied to the series and the mean and variance within that window were calculated. If the standardized mean or standardized variance of a certain window exceeds ± 3, a jump is identified, and the station is eliminated. Note that the discontinuity before and after 2013 is neglected in this process, because we have tackled this problem as described in Section 2.4.

3) If the ratio between 75th and 50th percentile is less than 1.07 or if the ratio between the 90th and 75th percentile is less than 1.1, we treat this station as the one with unreliably low visibilities and this station is eliminated.

After filtering with the above criteria, only the stations with sufficient valid samples (> 90 % availability) were selected for identifying SWHDs in this work as described in Section 2.4.

***3.*** *Section 2.3.1: The authors perform nested simulation with high resolution. However, the input meteorology data is still of low resolution. I wonder if this will affect the simulation results. Or is nested simulation really necessary?*

**Response:**

Sorry we didn't describe clearly in the previous version of manuscript. For the nested domain of Asia (11° S-55° N, 60°-150° E), the simulations had high horizontal resolution of 0.5° latitude by 0.625° longitude. Only the input lateral boundary conditions of tracer concentrations were taken from a global simulation at a lower resolution of 2° latitude by 2.5° longitude. The resolution of meteorological data that drove the chemical simulations was the same as that of the simulated chemical tracers.

***4.*** *Section 4.2: I suggest adding some discussion about the uncertainty of the trend of each species, as this can be significantly affected by uncertainties in emission (especially historical emission data) and chemistry processes in the model.*

**Response:**

We have added the following discussions in the second paragraph of Section 4.2:

"Note that these simulated trends of PM$_{2.5}$ components are of uncertainty, which can be influenced by the uncertainties in emission inventories of aerosols and aerosol precursors (Crippa et al., 2018; Li et al., 2017b; Zheng et al., 2018) as well as by the chemistry scheme in the model (Chen et al., 2019)."

5. *Section 5: When using equation (1) to make the partition, the contribution of each factor is assumed to be linear. This may not always be true. For example, both transport and PBL mixing can be affected by horizontal wind speed. So maybe a note is needed here when interpreting the results?*

**Response:**

Thanks for pointing it out. We have added the following sentence in Section 5:

"It should be noted that although process analysis is a helpful tool to quantify the contribution of each factor, it assumes linear contribution from each factor. This may not always be true because of the covariation of meteorological parameters."

6. *Figure 9 is very interesting and important. I wonder if the results are similar for different periods with different haze day trends.*

**Response:**

Figure 9 shows the process analysis for SWHDs in BTH for the whole period of 1985-2017. Following the reviewer's suggestion, we have done similar analyses for the two periods with different SWHD trends, 1992-2001 with a decreasing trend and 2003-2012 with an increasing trend, as shown in Figure S4. The results from these two periods (Fig. S4) are similar to those for 1985-2017 (Fig. 9), with transport, chemistry, cloud processes, dry deposition and PBL mixing contributed 70.2 % (67.2 %) , 14.5 % (16.2 %), -7.1 % (-8.3 %), -5.9 % (-5.9 %) and 2.3 % (2.3 %), respectively, to the SWHDs during 1992-2001 (2003-2012). The similarity can be explained by that the process analysis carried out for SWHDs during 1992-2001 (or 2003-2012) was compared to that for mean wintertime conditions of 1992-2001 (or 2003-2012). Therefore, the mechanism of the occurrence of SWHDs relative to the mean condition is the same, with transport process playing a dominant role. So, we present only the results obtained for the period of 1985-2017 in Fig. 9 of our manuscript.

[Figure]

**Figure S4.** Same as Figure 9 but for periods of **(a)** 1992-2001 and **(b)** 2003-2012.

**References:**

Chen, L., Zhu, J., Liao, H., Gao, Y., Qiu, Y., Zhang, M., and Li, N.: Assessing the formation and evolution mechanisms of severe haze pollution in Beijing−Tianjin−Hebei region by using process analysis, Atmos. Chem. Phys. Discuss., 2019, 1-42, 10.5194/acp-2019-245, 2019.

Crippa, M., Guizzardi, D., Muntean, M., Schaaf, E., Dentener, F., van Aardenne, J. A., Monni, S., Doering, U., Olivier, J. G. J., Pagliari, V., and Janssens-Maenhout, G.: Gridded emissions of air pollutants for the period 1970-2012 within EDGAR v4.3.2, Earth Syst. Sci. Data, 10, 1987-2013, 10.5194/essd-10-1987-2018, 2018.

He, J. J., Gong, S. L., Yu, Y., Yu, L. J., Wu, L., Mao, H. J., Song, C. B., Zhao, S. P., Liu, H. L., Li, X. Y., and Li, R. P.: Air pollution characteristics and their relation to meteorological conditions during 2014-2015 in major Chinese cities, Environ. Pollut., 223, 484-496, 10.1016/j.envpol.2017.01.050, 2017.

Li, J., Li, C. C., Zhao, C. S., and Su, T. N.: Changes in surface aerosol extinction trends over China during 1980-2013 inferred from quality-controlled visibility data, Geophysical Research Letters, 43, 8713-8719, 10.1002/2016gl070201, 2016.

Li, M., Zhang, Q., Kurokawa, J., Woo, J. H., He, K. B., Lu, Z. F., Ohara, T., Song, Y., Streets, D. G., Carmichael, G. R., Cheng, Y. F., Hong, C. P., Huo, H., Jiang, X. J., Kang, S. C., Liu, F., Su, H., and Zheng, B.: MIX: a mosaic Asian anthropogenic emission inventory under the international collaboration framework of the MICS-Asia and HTAP, Atmospheric Chemistry and Physics, 17, 935-963, 10.5194/acp-17-935-2017, 2017b.

Zheng, B., Tong, D., Li, M., Liu, F., Hong, C. P., Geng, G. N., Li, H. Y., Li, X., Peng, L. Q., Qi, J., Yan, L., Zhang, Y. X., Zhao, H. Y., Zheng, Y. X., He, K. B., and Zhang, Q.: Trends in China's anthropogenic emissions since 2010 as the consequence of clean air actions, Atmospheric Chemistry and Physics, 18, 14095-14111, 10.5194/acp-18-14095-2018, 2018.

**Thank you very much for your comments and suggestions.**

---

## Author Comment (AC2) · 22 Jul 2019

**Response to Comments of Reviewer #2**

**Manuscript number:** acp-2019-306

**Authors:** Ruijun Dang and Hong Liao

**Title:** Severe winter haze days in the Beijing-Tianjin-Hebei region from 1985-2017 and the roles of anthropogenic emissions and meteorology

**General comments:**

*The paper addresses the interannual variation of the severe winter haze days (SWHDs) in the Beijing-Tianjin-Hebei (BTH) region from 1985-2017 and the impacts of the anthropogenic emissions and meteorology on the variation. This study is of scientific importance and the research is well conducted. The paper is written with clear structure, good illustrations, and convincing discussions. The paper is subject to some issues described in the following. The authors are encouraged to consider these questions in their revision.*

**Response:**

Thanks to the reviewer for the helpful comments and suggestions. We have revised the manuscript carefully and the point-to-point responses are listed below.

**Specific Comments:**

1. *It appears that GEOS-Chem cannot capture the interannual variation of SWHDs intensity well (see Figure 4f). In Figure 4f, R is missing. Is R statistically significant? How does this uncertainty affect the results in Figures 6, 11 and 12 and the associated discussion in the text?*

**Response:**

Thanks for pointing it out. Inspired by your comment #3, we realized that the simulated values of SWHD intensity in the previous version of manuscript were unadjusted, which were incompatible with the simulated SWHD frequencies which were adjusted according to the model biases. To address this issue, the intensity is now calculated as the average of daily mean $PM_{2.5}$ concentrations over all SWHDs during a winter by subtracting the mean bias (MB, as described in Section 2.4) from the original simulations. For example, for the grid at Baoding where the simulated wintertime $PM_{2.5}$ has a MB of -28.1 $\mu g\ m^{-3}$, the intensity of simulated SWHDs is adjusted as 249.6 $\mu g\ m^{-3}$ (221.5 $\mu g\ m^{-3}$ minus -28.1 $\mu g\ m^{-3}$) over 2013-2017. We have revised the related figures (Figure 4, 6, 8, 11 and 12) and descriptions in the text accordingly. In the revised Figure 4f, the correlation coefficient R between simulated and observed SWHD intensity is

0.58. The adjustment has a small impact on the results of simulated SWHD intensity in BTH (Figure 11) because the model has a small MB of 4.5 μg m$^{-3}$ in simulated PM$_{2.5}$ concentrations over BTH.

2. *In the simulation experiments, biomass burning emissions are interannually variable from 1997-2016. The interannual variation of biomass burning is large globally and regionally, which may not be ignored. Therefore, the EMIS simulation may be driven by the interannual variations in both anthropogenic and biomass burning emissions, while the MET simulation is influenced by both biomass burning emissions and meteorology. Please explain.*

**Response:**

Sorry for the confusion. We did not describe clearly our treatment of biomass burning in the previous version of manuscript. The CTRL simulation considers variations in meteorological parameters, anthropogenic emissions, and biomass burning emissions over 1985-2017. In MET simulation, meteorological parameters are allowed to vary from 1985 to 2017, but anthropogenic and biomass burning emissions are fixed at the year 2015 levels. The MET simulation thus represents the impact of variations in meteorological parameters on the variations of SWHDs. In EMIS simulation, anthropogenic and biomass burning emissions are allowed to vary over 1985-2017, while meteorological parameters are fixed at year 1985 levels. The related descriptions in Section 2.3.3 have been clarified in the revised manuscript.

Even though the EMIS simulation includes variations in both anthropogenic and biomass burning emissions, the impact of biomass burning on SWHDs in BTH is insignificant. Large biomass burning events often occur in Southeast Asia and Russia in spring, which are reported to have significant impact on the interannual variations of air pollution in spring in the adjacent areas such as southern Yunnan province, northern Heilongjiang province, as well as part of Inner Mongolia. For other parts of China during winter, the impact of biomass burning is small (Mao et al., 2016; Lou et al., 2015).

3. *In the abstract, the authors stated: "the correlation coefficient between the simulated and observed SWHDs is 0.98 at 161 grids in China". This claim is based on Figure 4 that compares simulated and observed SWHD in terms of frequency and intensity. In section 2.4, the authors defined SWHDs for the observations and simulations. The simulated SWHD is adjusted according to the simulation biases. It is not clear if the simulated results presented in Figure 4 are the original simulations or adjusted values. If they are adjusted values, it should be stated there. Also, the claim in the abstract should be revised accordingly.*

**Response:**

The model results presented in the revised Figure 4 are adjusted values (see the response to your comment #1 for details).

To clarify, we have added the following sentence in the abstract: "Observed SWHDs were defined as the days with daily mean $PM_{2.5}$ concentration exceeding 150 μg m$^{-3}$, and simulated SWHDs were identified by using the same threshold but with adjustment on the basis of simulation biases."

We have also added the following sentence in Section 3.2 when we present Figure 4: "Note that, for each grid, the simulated intensity was an adjusted value according to the model mean bias (MB) calculated in Section 2.4. For example, for the grid at Baoding where the simulated wintertime $PM_{2.5}$ has a MB of -28.1 μg m$^{-3}$, the intensity of simulated SWHDs is adjusted as 249.6 μg m$^{-3}$ (221.5 μg m$^{-3}$ minus -28.1 μg m$^{-3}$) over 2013-2017."

4. *In generally, a moving average tends to smooth year-to-year variation. But in Figure 12, the author stated that using a "9-year weighted moving average" can reserve interannual variations in the SWHDs frequency and intensity. I wonder how this works. Why 9 years? Which climatic influence did the authors try to remove? Why to remove the fluctuations of more than 9 years?*

**Response:**

To extract the interannual component from the original timeseries, 7-10 years high pass Lanczos filters (Duchon et al., 1979) have been widely applied in previous studies (Salinger et al., 2001; Wu et al., 2010; Schneider et al., 2012; Chen et al., 2015; Wang et al., 2016; Liu et al., 2017; Sun and Wang, 2018; Wang et al., 2019), and among them the 9-year high pass Lanczos filter is the most common one. This filter can effectively remove the decadal variabilities of low frequency while preserve the interannual signals of high frequency. Therefore, in Figure 12 of this study, we selected the 9-year high pass Lanczos filter for obtaining interannual variations of SWHD timeseries.

5. *In the title, the term of "meteorological parameters" is not suitable. It is better to use term of "meteorology", "meteorological factors", or "meteorological variables"?*

**Response:**

We have replaced "meteorological parameters" by "meteorology" in the title.

6. *Figure 8, it is hard to identify a tick for a year in the x-axis.*

**Response:**

To make it clear, we have modified the length of tick marks in x-axis. The revised Figure 8 is displayed below.

[Figure]

**Figure 8.** Simulated temporal variations in frequency (days, black line), intensity (μg m⁻³, red line) and concentrations of PM$_{2.5}$ components (μg m⁻³, bars): ammonia (blue), nitrate (yellow), sulfate (red), black carbon (purple) and organic aerosols (green) of regional SWHDs in BTH from 1985-2017. Also shown are linear trends (dashed lines) for frequency and intensity, which are statistically significant above the 95 % confidence level.

*7. P. 7, Line 32, replace "horizontal" with "spatial".*

**Response:**

Replaced.

*8. In Author contributions, delete "from all coauthors".*

**Response:**

Deleted.

**References:**

Chen, S. F., Wu, R. G., and Chen, W.: The Changing Relationship between Interannual Variations of the North Atlantic Oscillation and Northern Tropical Atlantic SST, J. Clim., 28, 485-504, 10.1175/jcli-d-14-00422.1, 2015.

Duchon, C. E.: LANCZOS FILTERING IN ONE AND 2 DIMENSIONS, J. Appl. Meteorol., 18, 1016-1022, 10.1175/1520-0450(1979)018<1016:lfioat>2.0.co;2, 1979.

Liu, G., Zhao, P., and Chen, J.: Possible Effect of the Thermal Condition of the Tibetan Plateau on the Interannual Variability of the Summer Asian–Pacific Oscillation, J. Clim., 30, 9965-9977, 10.1175/jcli-d-17-0079.1, 2017.

Lou, S. J., Liao, H., Yang, Y., and Mu, Q.: Simulation of the interannual variations of tropospheric ozone over China: Roles of variations in meteorological parameters and anthropogenic emissions, Atmos. Environ., 122, 839-851, 10.1016/j.atmosenv.2015.08.081, 2015.

Mao, Y. H., Liao, H., Han, Y. M., and Cao, J. J.: Impacts of meteorological parameters and emissions on decadal and interannual variations of black carbon in China for 1980-2010, J. Geophys. Res.-Atmos.,

121, 1822-1843, 10.1002/2015jd024019, 2016

Salinger, M. J., Renwick, J. A., and Mullan, A. B.: Interdecadal Pacific Oscillation and South Pacific climate, International Journal of Climatology, 21, 1705-1721, 10.1002/joc.691, 2001.

Schneider, D. P., Okumura, Y., and Deser, C.: Observed Antarctic Interannual Climate Variability and Tropical Linkages, J. Clim., 25, 4048-4066, 10.1175/jcli-d-11-00273.1, 2012.

Sun, B., and Wang, H. J.: Interannual Variation of the Spring and Summer Precipitation over the Three River Source Region in China and the Associated Regimes, J. Clim., 31, 7441-7457, 10.1175/jcli-d-17-0680.1, 2018.

Wang, J., Zeng, N., and Wang, M. R.: Interannual variability of the atmospheric $CO_2$ growth rate: roles of precipitation and temperature, Biogeosciences, 13, 2339-2352, 10.5194/bg-13-2339-2016, 2016.

Wang, M., Jia, X. J., Ge, J. W., and Qian, Q. F.: Changes in the Relationship Between the Interannual Variation of Eurasian Snow Cover and Spring SAT Over Eastern Eurasia, J. Geophys. Res.-Atmos., 124, 468-487, 10.1029/2018jd029077, 2019.

Wu, B., Li, T., and Zhou, T. J.: Relative Contributions of the Indian Ocean and Local SST Anomalies to the Maintenance of the Western North Pacific Anomalous Anticyclone during the El Nino Decaying Summer, J. Clim., 23, 2974-2986, 10.1175/2010jcli3300.1, 2010.

**Thank you very much for your comments and suggestions.**

---

## Author Response (AR1)

**Response to Comments of Reviewer #1**

**Manuscript number:** acp-2019-306

**Authors:** Ruijun Dang and Hong Liao

**Title:** Severe winter haze days in the Beijing-Tianjin-Hebei region from 1985-2017 and the roles of anthropogenic emissions and meteorology

**General comments:**

*This paper investigates the roles of meteorology and emission on the long-term changes of winter haze in the BTH region. To my knowledge, this is the first attempt to quantify these two effects for historical periods. The results shown here thus have important implications for the understanding of haze formation and air quality control in north China. The paper is also well organized and easy to follow. I only have a few minor comments as listed below.*

**Response:**

Thanks to the reviewer for the helpful comments and suggestions. We have revised the manuscript carefully and the point-to-point responses are listed below.

**Specific Comments:**

*1.  It is better to include some more quantitative results in the abstract, such as those in Table 1 and Figure 12.*

**Response:**

We have added the following quantitative descriptions in the abstract to summarize the major results from Table 1 and Figure 12:

"Process analysis on all SWHDs during 1985-2017 indicated that transport was the most important process for the formation of SWHDs in BTH with a relative contribution of 65.3 %, followed by chemistry (17.6%), cloud processes (-7.5%), dry deposition (-6.4%) and PBL mixing (3.2%). Further examination showed that SWHDs exhibited large interannual variations in frequency and intensity, which were mainly driven by changes in meteorology."

*2.  Historical visibility data usually have higher uncertainty and noise level, I wonder if any quality control is enforced?*

**Response:**

Yes, quality control was enforced in this work. First, the downloaded daily visibility

data from National Climatic Data Center (NCDC) have already undergone extensive quality control (see ftp://ftp.ncdc.noaa.gov/pub/data/gsod/readme.text and ftp://ftp.ncdc.noaa.gov/pub/data/noaa/ish-qc.pdf for details), which to a large extent have eliminated many random errors in the original data.

Second, to ensure the consistency and continuity of the downloaded data, further quality control was carried out to remove unreliable outliers/stations following the methods in previous studies (Li et al., 2016; He et al., 2017). For each station:

1) If the standardized daily visibility ($V_i$) exceeds ± 4, the daily data is then marked as outlier and removed from the original time series.

2) A 3-year sliding window was applied to the series and the mean and variance within that window were calculated. If the standardized mean or standardized variance of a certain window exceeds ± 3, a jump is identified, and the station is eliminated. Note that the discontinuity before and after 2013 is neglected in this process, because we have tackled this problem as described in Section 2.4.

3) If the ratio between 75th and 50th percentile is less than 1.07 or if the ratio between the 90th and 75th percentile is less than 1.1, we treat this station as the one with unreliably low visibilities and this station is eliminated.

After filtering with the above criteria, only the stations with sufficient valid samples (> 90 % availability) were selected for identifying SWHDs in this work as described in Section 2.4.

3. *Section 2.3.1: The authors perform nested simulation with high resolution. However, the input meteorology data is still of low resolution. I wonder if this will affect the simulation results. Or is nested simulation really necessary?*

**Response:**

Sorry we didn't describe clearly in the previous version of manuscript. For the nested domain of Asia (11° S-55° N, 60°-150° E), the simulations had high horizontal resolution of 0.5° latitude by 0.625° longitude. Only the input lateral boundary conditions of tracer concentrations were taken from a global simulation at a lower resolution of 2° latitude by 2.5° longitude. The resolution of meteorological data that drove the chemical simulations was the same as that of the simulated chemical tracers.

4. *Section 4.2: I suggest adding some discussion about the uncertainty of the trend of each species, as this can be significantly affected by uncertainties in emission (especially historical emission data) and chemistry processes in the model.*

**Response:**

We have added the following discussions in the second paragraph of Section 4.2:

"Note that these simulated trends of $PM_{2.5}$ components are of uncertainty, which can be influenced by the uncertainties in emission inventories of aerosols and aerosol precursors (Crippa et al., 2018; Li et al., 2017b; Zheng et al., 2018) as well as by the chemistry scheme in the model (Chen et al., 2019)."

5. *Section 5: When using equation (1) to make the partition, the contribution of each factor is assumed to be linear. This may not always be true. For example, both transport and PBL mixing can be affected by horizontal wind speed. So maybe a note is needed here when interpreting the results?*

**Response:**

Thanks for pointing it out. We have added the following sentence in Section 5:

"It should be noted that although process analysis is a helpful tool to quantify the contribution of each factor, it assumes linear contribution from each factor. This may not always be true because of the covariation of meteorological parameters."

6. *Figure 9 is very interesting and important. I wonder if the results are similar for different periods with different haze day trends.*

**Response:**

Figure 9 shows the process analysis for SWHDs in BTH for the whole period of 1985-2017. Following the reviewer's suggestion, we have done similar analyses for the two periods with different SWHD trends, 1992-2001 with a decreasing trend and 2003-2012 with an increasing trend, as shown in Figure S4. The results from these two periods (Fig. S4) are similar to those for 1985-2017 (Fig. 9), with transport, chemistry, cloud processes, dry deposition and PBL mixing contributed 70.2 % (67.2 %) , 14.5 % (16.2 %), -7.1 % (-8.3 %), -5.9 % (-5.9 %) and 2.3 % (2.3 %), respectively, to the SWHDs during 1992-2001 (2003-2012). The similarity can be explained by that the process analysis carried out for SWHDs during 1992-2001 (or 2003-2012) was compared to that for mean wintertime conditions of 1992-2001 (or 2003-2012). Therefore, the mechanism of the occurrence of SWHDs relative to the mean condition is the same, with transport process playing a dominant role. So, we present only the results obtained for the period of 1985-2017 in Fig. 9 of our manuscript.

[Figure]

**Figure S4.** Same as Figure 9 but for periods of **(a)** 1992-2001 and **(b)** 2003-2012.


**Response:**

To make it clear, we have modified the length of tick marks in x-axis. The revised Figure 8 is displayed below.

[Figure]

**Figure 8.** Simulated temporal variations in frequency (days, black line), intensity (μg m⁻³, red line) and concentrations of PM₂.₅ components (μg m⁻³, bars): ammonia (blue), nitrate (yellow), sulfate (red), black carbon (purple) and organic aerosols (green) of regional SWHDs in BTH from 1985-2017. Also shown are linear trends (dashed lines) for frequency and intensity, which are statistically significant above the 95 % confidence level.

*7. P. 7, Line 32, replace "horizontal" with "spatial".*

**Response:**

Replaced.

*8. In Author contributions, delete "from all coauthors".*

**Response:**

Deleted.

*Correspondence to*: Hong Liao (hongliao@nuist.edu.cn)

**Abstract:** We applied a global 3-D chemical transport model (GEOS-Chem) to examine the variations in the frequency and intensity in severe winter haze days (SWHDs) in BTH from 1985-2017 and quantified the roles of changes in anthropogenic emissions and/or meteorological parameters. Observed SWHDs were defined as the days with daily mean $PM_{2.5}$ concentration exceeding 150 μg m$^{-3}$, and simulated SWHDs were identified by using the same threshold but with adjustment on the basis of simulation biases. Comparisons between the simulated SWHDs and those obtained from the observed $PM_{2.5}$ concentrations and atmospheric visibility showed that the model can capture the spatial and temporal variations of SWHDs in China; the correlation coefficient between the simulated and observed SWHDs is 0.98 at 161 grids in China. From 1985-2017, with changes in both anthropogenic emissions and meteorological parameters, the simulated frequency (total severe haze days in winter) and intensity ($PM_{2.5}$ concentration averaged over severe haze days in winter) of SWHDs in BTH showed increasing trends of 4.5 days decade$^{-1}$ and 13.5 μg m$^{-3}$ decade$^{-1}$, respectively. The simulated frequency exhibited fluctuations from 1985-2017, with a sudden decrease from 1992-2001 (29 days to 10 days) and a rapid growth from 2003-2012 (16 days to 47 days). The sensitivity simulations indicated that variations in meteorological parameters played a dominant role during 1992-2001, while variations in both emissions and meteorological parameters were important for the simulated frequency trend during 2003-2012 (simulated trends were 27.3 days decade$^{-1}$ and 12.5 days decade$^{-1}$ owing to changes in emissions alone and changes in meteorology alone, respectively). The simulated intensity showed a steady increase from 1985-2017, which was driven by changes in both emissions and meteorology. Process analysis on all SWHDs during 1985-2017 indicated that transport was the most important process for the formation of SWHDs in BTH with a relative contribution of 65.3 %, followed by chemistry (17.6%), cloud processes (-7.5%), dry deposition (-6.4%) and PBL mixing (3.2%). Further examination showed that SWHDs exhibited large interannual variations in frequency and intensity,

[revised manuscript text omitted]
. 1). Note that these simulated trends of PM$_{2.5}$ components are of uncertainty, which can be influenced by the uncertainties in emission inventories of aerosols and aerosol precursors (Crippa et al., 2018; Li et al., 2017b; Zheng et al., 2018) as well as by the chemistry scheme in the model (Chen et al., 2019). The simulations in this study did not include secondary organic aerosols; therefore, the concentrations of organic aerosols  might have been underestimated. Previous studies also showed that the GEOS-Chem model tends to overestimate nitrate and underestimate sulfate in winter (Wang et al., 2013).

**5 Key processes for the occurrence of SWHDs over BTH**

[revised manuscript text omitted]